# Stabilisation of HIF signalling in the mouse epicardium extends embryonic potential and neonatal heart regeneration

Elisabetta Gamen[1,2], Eleanor L Price[2], Daniela Pezzolla[3], Carla De Villiers[1,2], Mala Gunadasa-Rohling[1,2], Adam B Lokman[1,2], Maria-Alexa Cosma[1,2], Judith Sayers[1,2], Carolina Roque Silva[1,2], Rafik Salama[4], David Robert Mole[4], Tammie Bishop[2,4,5], Chris W Pugh[6], Robin P Choudhury[3], Carolyn A Carr[2], Joaquim Miguel Vieira[7]*[†], Paul R Riley[1,2]*[†]

[1]Institute of Developmental and Regenerative Medicine (IDRM), Oxford, United Kingdom; [2]Burdon-Sanderson Cardiac Science Centre, Department of Physiology, Anatomy and Genetics, University of Oxford, Oxford, United Kingdom; [3]Division of Cardiovascular Medicine, Radcliffe Department of Medicine, University of Oxford, Oxford, United Kingdom; [4]NDM Research Building, University of Oxford, Oxford, United Kingdom; [5]Ludwig institute for Cancer Research, University of Oxford, Oxford, United Kingdom; [6]Target Discovery Institute, University of Oxford, Oxford, United Kingdom; [7]School of Cardiovascular and Metabolic Medicine and Sciences, British Heart Foundation Centre of Research Excellence, King's College London, London, United Kingdom

*For correspondence:
joaquim.nunes_vieira@kcl.ac.uk
(JMV);
paul.riley@dpag.ox.ac.uk (PRR)

[†]These authors contributed equally to this work

Competing interest: The authors declare that no competing interests exist.

## eLife Assessment

This **valuable** study investigates the role of HIF1a signaling in epicardial activation and neonatal heart regeneration in mice. Using a combination of genetic and pharmacological approaches, the authors demonstrate that stabilization of HIF1a enhances epicardial activation and extends the regenerative capacity of the heart beyond the typical neonatal window following myocardial infarction. The main conclusion is well supported by **solid** data, although some minor concerns regarding experimental interpretation require further clarification to ensure accuracy.

**Abstract** In humans, new-born infants can regenerate their heart during early life. This is modelled in the mouse, where regenerative capacity is maintained for the first week after birth but lost thereafter. Reactivation of this process holds great therapeutic potential; however, the molecular pathways that might be targeted to extend neonatal regeneration remain elusive. Here, we explored a role for hypoxia and HIF signalling on the regulation of epicardial activity in the developing mouse heart and in modulating the response to injury. Hypoxic regions were found in the epicardium from mid-gestation, associating with HIF-1α and HIF-2α, and expression of the epicardial master regulator Wilms' tumour 1 (WT1). Epicardial deletion of *Hif1α* reduced WT1 levels, leading to impaired coronary vasculature. Targeting of the HIF degradation enzyme PHD, through pharmacological inhibition with a clinically approved drug or epicardial-specific genetic deletion of *Egln1*, stabilised HIF and promoted WT1 activity ex vivo. Finally, a combination of genetic and pharmacological stabilisation of HIF during neonatal heart injury led to prolonged epicardial activation, preservation of myocardium, augmented infarct resolution and preserved function beyond the 7-day regenerative window. These findings suggest modulation of HIF signalling extends epicardial activation to maintain

myocardial survival beyond the neonatal regenerative window and may represent a viable strategy for treating ischaemic heart disease.

## Introduction

Unlike the adult mammalian heart, the neonatal mouse heart exhibits a remarkable regenerative capacity after myocardial infarction (MI) up to 7 days after birth (*Aaboud et al., 2018*). A similar regenerative capacity was shown in new-born humans (*Haubner et al., 2016*). This regenerative potential is linked to the epicardium, a mesothelial layer covering the outer surface of the heart. During development, a subset of epicardial cells undergoes epithelial to mesenchyme transition (EMT), generating epicardium-derived cells (EPDCs) that contribute to coronary vessel formation and cardiomyocyte proliferation and compaction (*Simões and Riley, 2018*). A number of transcription factors are expressed in the developing epicardium, including Wilms' tumour 1 (WT1), which is instrumental in the regulation of EMT. Loss of WT1 impairs the ability of EPDCs to migrate into the underlying sub-epicardial region and mutant embryos die between E14.5 and E16.5 due to cardiovascular failure (*Martínez-Estrada et al., 2010*). EPDC plasticity gradually decreases during embryogenesis and after birth, the epicardium becomes quiescent. Embryonic genes, such as *Wt1*, are downregulated and EMT ceases (*Zhou and Pu, 2011*).

Following injury, the epicardium reactivates, resembling an embryonic-like state. Epicardial cells re-express WT1 (*Wang et al., 2015*), proliferate, undergo EMT, and differentiate into fibroblasts to stimulate tissue repair (*Zhou et al., 2011*). Thymosin β4 (Tβ4) pre-treatment results in EPDC-mediated neovascularisation after injury (*Smart et al., 2007a*) and induces limited de novo cardiomyocyte formation by differentiation from a progenitor population of epicardial origin (*Smart et al., 2011*).

Hypoxia, or low oxygen levels, has been linked to tissue regeneration (*Price et al., 2019*), and hypoxia-inducible factors (HIFs) play a central role in cellular responses to hypoxia (*Kaelin and Ratcliffe, 2008*). HIFs are heterodimeric transcription factors, consisting of α and β subunits (*Dunwoodie, 2009*), with the latter being constitutively expressed, whereas expression of the α subunit is oxygen-dependent (reviewed in *Lee et al., 2019*). Under well-oxygenated conditions (i.e. normoxia), prolyl hydroxylase domain proteins (PHD1-3) hydroxylate specific conserved proline residues on the α subunit, tagging it for proteasomal degradation. When oxygen levels drop, PHD activity is inhibited, enabling the α subunit to translocate to the nucleus and bind to hypoxia response elements (HRE) regulating transcription of a plethora of target genes (*Appelhoff et al., 2004*). In humans, mice, and rats, there are three isoforms of the α subunit, but the best characterised are HIF-1α and HIF-2α activating distinct, although partially overlapping, sets of genes (*Smythies et al., 2019*). Hypoxia and HIF activation are evident throughout the developing heart (*Dunwoodie, 2009*) and disruption of oxygen sensing pathways during development results in a wide range of cardiac abnormalities (*Romanowicz et al., 2021*). Notably, WT1 expression is upregulated in the heart and kidneys of rats exposed to hypoxia and an HRE binding site for HIF-1α has been identified within the *Wt1* promoter (*Wagner et al., 2003*) suggesting a link between hypoxia, HIF signalling, and epicardial activation. In this context, the role of epicardial HIF signalling in heart development and regeneration remains unclear. Studies have shown that stabilising HIF-1α inhibits EPDC migration in avian embryos (*Tao et al., 2013*) but promotes their differentiation into vascular smooth muscle cells (VSMCs; *Tao et al., 2018*). Endothelium-specific depletion of *Egln1* and *Eglin3* (encoding PHD2 and PHD3, respectively) promotes cardiomyocyte proliferation and prevents left ventricular failure in a model of MI (*Fan et al., 2019*), and exposure to hypoxia following MI induces a regenerative response in adult mice by metabolic reprogramming of cardiomyocytes leading to cell cycle re-entry (*Nakada et al., 2017*).

The epicardium and subepicardial space are identified as hypoxic niches, housing a unique progenitor cell population (*Kocabas et al., 2012*; *Hesse et al., 2021*). Given that hypoxia has been implicated in regenerative responses post-MI, and epicardial WT1 expression is HIF dependent, we hypothesised that HIF signalling contributes to epicardial cell activation during heart development and following injury and as such might represent a therapeutic target for extending the regenerative window (*Porrello et al., 2011*). Here, we demonstrate that the developing epicardium is hypoxic and epicardial-specific deletion of *Hif1a* disrupts epicardial EMT and heart vascularisation. In mice, we observe a gradual and concurrent decrease in WT1 expression and HIF-mediated signalling over the first week of life. Stabilisation of HIF signalling was sufficient to maintain activation of the epicardium

beyond postnatal day (P) 7, improve cardiac remodelling and preserve function post-MI. These findings provide novel insight into the molecular mechanism regulating epicardial activation in the mouse heart and suggest that modulation of HIF signalling in the dormant epicardium may enhance cardiac repair and regeneration, representing a potential therapeutic strategy for the treatment of ischaemic heart disease and heart failure.

## Results

### The epicardium is hypoxic during development

To study the role of low oxygen tension in the developing heart, we initially investigated the physiological levels of hypoxia in situ from mid- to late gestation stages, when the epicardium forms and epicardial EMT takes place. Following in utero treatment with the marker Pimonidazole-HCl (Hypoxyprobe-1, HP1; *Raleigh et al., 1999*), we probed embryonic hearts at various developmental time points, ranging from E12.5 to E18.5 using an anti-pimonidazole fluorescence-conjugated monoclonal antibody (HP1), alongside immunostaining for the spatiotemporal expression of HIF-1α and HIF-2α.

At E12.5, characterised by a well-established epicardial layer amidst highly trabeculated myocardium, we observed extensive regions of hypoxia, mostly in areas of dense myocardium, such as the compact wall of the atrioventricular (AV) groove (*Figure 1a*). Notably, HP1 staining co-localised with the epicardial marker WT1 (*Figure 1a*), and both HIF-1α (*Figure 1—figure supplement 1a*) and HIF-2α at the apex of the heart. Additionally, strong HIF-1α signals were noted at the base of the heart (*Figure 1a*), while HIF-2α expression extended into the myocardial compartment (*Figure 1—figure supplement 2a*).

By E14.5, extensive HP1 staining persisted in areas of thicker compact myocardium, such as the interventricular septum (IVS; *Figure 1b*). Co-localisation with WT1 and HIF-1α was observed predominantly in the epicardium, notably in the apex (*Figure 1—figure supplement 1b*), and base regions. While HIF-2α co-localisation with HP1 was present, it was primarily within the myocardium (*Figure 1b*, *Figure 1—figure supplement 2b*).

Analysis of hearts at E16.5 and E18.5 revealed a progressive decrease in HP1 signal, becoming largely confined to WT1-expressing epicardial cells at later stages (*Figure 1c and d*). Similarly, HIF-1α localisation was restricted in discrete regions of the outermost layer of the heart (*Figure 1—figure supplement 1c, d*), whereas HIF-2α expression transitioned from the epicardium (notably at the apex of the heart) to some sporadic expression in underlying cardiomyocytes by E18.5. (*Figure 1—figure supplement 2c, d*, *Figure 1c and d*). Collectively, these data suggest that WT1+ cells making up the epicardial layer of the developing heart become hypoxic from mid-gestation onwards, preferentially expressing HIF-1α, whereas HIF-2α expression predominantly occurs within the forming myocardium.

### HIF-1α epicardial deletion reduces the number of Wt1+ cells and alters coronary vessel development in E16.5 embryos

Given that the epicardium is hypoxic during mid-late gestation, and we observed expression of HIF-1α by WT1+ cells within the epicardium, we next determined the functional implications of HIF-1α on epicardial development. To specifically target *Hif1α* expression in the epicardium, we crossed the tamoxifen-inducible epicardial-specific *CreERT2* driver, *Wt1*<sup>CreERT2/+</sup> (*Zhou et al., 2008*) with mice in which exon 2 of the *Hif1a* gene is flanked by LoxP sites (*Hif1a*<sup>fl/fl</sup>). Importantly, there is no known phenotype attributed to hemizygosity for Wt1 (*Moore et al., 1999*) as might be associated with the *Wt1*<sup>CreERT2/+</sup> mice (*Zhou et al., 2008*) and as such any interpretation of downstream effect is attributed to epicardial loss of HIF-1α. Pregnant *Wt1*<sup>CreERT2/+</sup>;*Hif1a*<sup>fl/fl</sup> females were injected with tamoxifen at E9.5 and E10.5, to target epicardial development, and embryos were collected at E16.5. Immunostaining for WT1 and the epicardial marker podoplanin (PDPN) revealed a marked reduction in the number of WT1+ cells in *Wt1*<sup>CreERT2/+</sup>;*Hif1a*<sup>fl/fl</sup> mutant (KO) hearts as compared to littermate controls (CTR; *Figure 2a and b*), affecting both the epicardial layer (*Figure 2c*; CTR: 0.72±0.11; KO: 0.32±0.010; p=0.0268) and the myocardium (*Figure 2d*; CTR: 3.96±1.01; KO: 0.98±0.25; p=0.0461). Notably, deletion of *Hif1a* did not affect HIF-2α expression in WT1+ cells (*Figure 2—figure supplement 1a*). Reduction in the number of Wt1+ cells in the myocardial compartment suggested an impairment in epicardial EMT. Hearts were also probed for the endothelial marker endomucin (EMCN), which also marks the endocardium and the outline of myocardial trabeculae. No significant alterations were

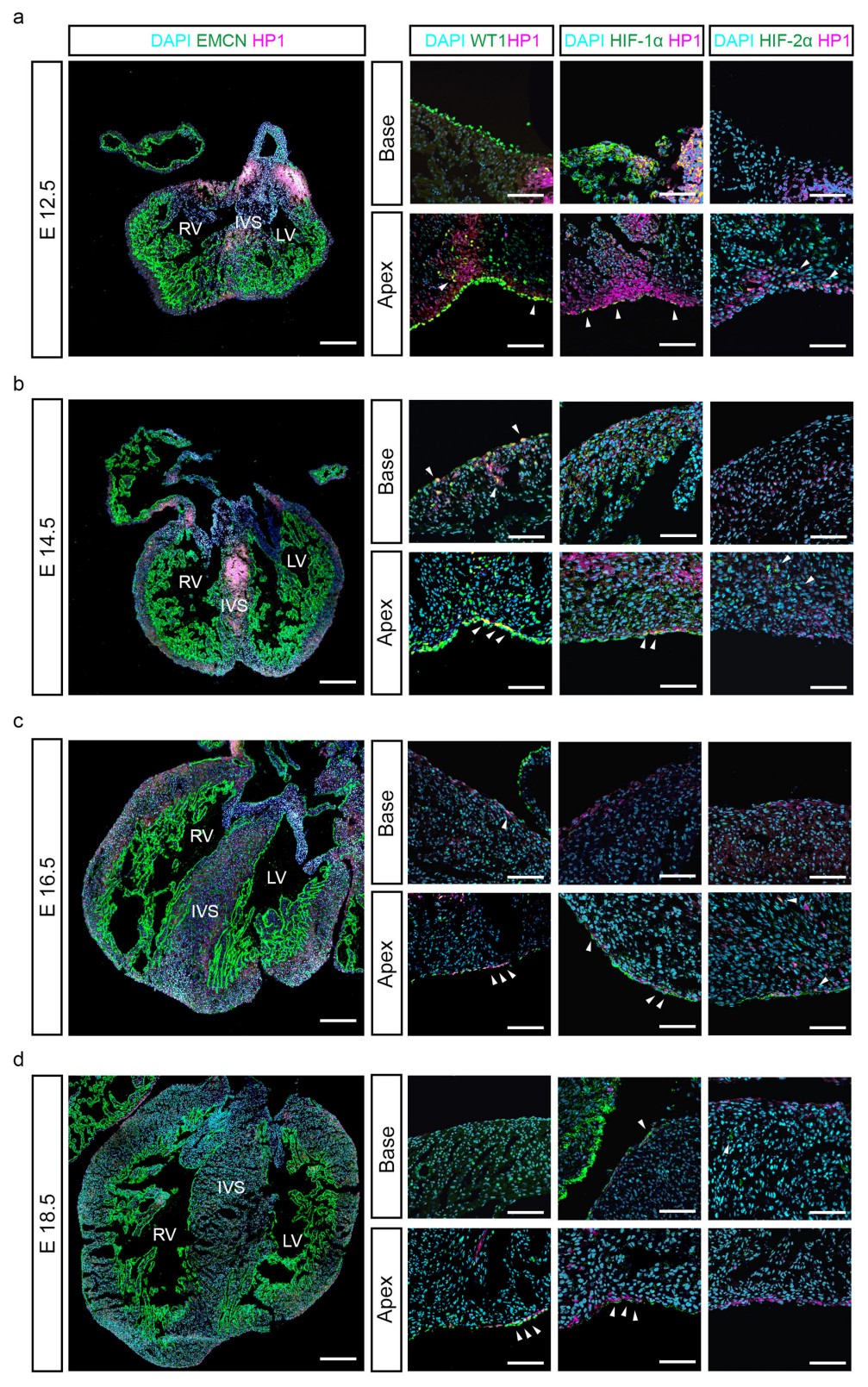

**Figure 1.** The epicardium is hypoxic at later stages of heart development. Representative images of immunostaining for DAPI (cyan), HP1 (magenta) and EMCN, WT1, HIF-1α and HIF-2α (green) on serial cryosections of foetal hearts at E12.5 (**a**), E14.5 (**b**), E16.5 (**c**), and E18.5 (**d**). Arrowheads indicate overlap between HP1 and

*Figure 1 continued on next page*

*Figure 1 continued*

WT1, HIF-1α or HIF-2α. n=6 hearts per stage. IVS = interventricular septum, RV = right ventricle, LV = left ventricle. Whole heart scale bars, 200 μm; high-magnification scale bars, 100 μm.

The online version of this article includes the following figure supplement(s) for figure 1:

**Figure supplement 1.** HIF-1α is expressed in the epicardium at different stages of heart development.

**Figure supplement 2.** HIF-2α is expressed in the myocardium at different stages of heart development.

---

observed in myocardial compaction (*Figure 2e*), myocardial trabeculation as assessed by fractal analysis (*Figure 2f*), or the proliferation rate of WT1$^+$ cells between mutants and controls (*Figure 2—figure supplement 1c, d*).

Given the essential role of the epicardium in coronary vessel development (*Smart et al., 2007b*), we assessed coronary vessel formation using whole-mount immunofluorescence staining for EMCN associated with AngioTool software quantification (*Zudaire et al., 2011*). Mutant hearts exhibited significantly impaired coronary vasculature, as evidenced by decreased total vessel length (microns; *Figure 2h*, CTR: 129229±5509; KO: 81073±10915; p=0.0170), number of junctions (*Figure 2i*, Mean ± SEM; CTR: 2505±63.74; KO: 1457±282.1; *P*=0.0223) and vessel end points (*Figure 2j*, Mean ± SEM; CTR: 3422±249.5; KO: 2429±218.6; p=0.0402).

During development, EPDCs undergo EMT and migrate into the sub-epicardial space to support coronary vessel formation. Thus, we sought to investigate whether the vascular impairment observed in the *Wt1$^{CreERT2/+}$;Hif1a$^{fl/fl}$* mutant embryos was due to compromised epicardial EMT. Epicardial explants from *Wt1$^{CreERT2/+}$;Hif1a$^{fl/fl}$* embryos treated with tamoxifen showed a reduced HIF-1α expression (*Figure 2—figure supplement 2b*; CTR: 0.99±0.16; KO: 0.46±0.12; p=0.0397) without affecting HIF-2α levels (*Figure 2—figure supplement 2c*; CTR: 2505±63.74; KO: 1457±282.1; p=0.223). Tamoxifen-treated epicardial explants from *Wt1$^{CreERT2/+}$;Hif1a$^{fl/fl}$* hearts revealed an increased epithelial and reduced mesenchymal phenotype. This was evident by reduced stress fibre formation (*Figure 2k*), reduced alpha-smooth muscle actin (α-SMA) staining (*Figure 2l*), and altered sub-cellular localisation of the EMT marker and structural component of cell-cell adhesion complexes zonula occludens-1 (ZO-1) (*Figure 2m*) compared to controls. Specifically, mutant explants showed a predominant membrane α-SMA staining, demarcating their epithelial cell shape, whereas control cells displayed a more intense cytoplasmic labelling of the filamentous actin cytoskeleton (*Figure 2l*). Similarly, in control cells, the localisation ZO-1 protein was predominantly cytoplasmic and/or nuclear, whereas in mutant cells, ZO-1 staining was membranous, indicating a lesser migratory and more epithelial-like phenotype (*Figure 2m*). Consistent with our in vivo findings, explants derived from *Wt1$^{CreERT2/+}$;Hif1α$^{fl/fl}$* embryos exhibited decreased WT1 expression, indicating impaired epicardial EMT (*Figure 2n and o*; CTR: 725.7±86.93; KO: 490.9±37.65; p=0.0479).

To elucidate the direct regulation of WT1 by HIF-1α, we interrogated a publicly available dataset of chromatin immunoprecipitation coupled with next generation sequencing (ChIP-seq) generated in the renal adenocarcinoma 786-O cell line (*Smythies et al., 2019*). ChIP-seq analysis confirmed direct HIF binding within intron 3 of the *Wt1* gene (*Figure 2—figure supplement 3*), consistent with previous findings (*Wagner et al., 2003*).

In summary, epicardial-specific deletion of *Hif1a* in vivo and ex vivo significantly reduced the number of WT1+ cells, both in the epicardium and underlying myocardium, arising through EMT affecting coronary vessel development at E16.5. Taken together, our data demonstrate that HIF regulates the expression of WT1 and associated epicardial EMT.

## Stabilisation of HIF signalling enhances Wt1 expression and epicardial EMT

We next sought to investigate whether stabilising HIF signalling under normoxic conditions is sufficient to enhance epicardial EMT. To this end, we established epicardial explants from *Rosa26$^{+/CreERT2}$;Egln1$^{fl/fl}$* mice, where Cre-mediated recombination induced loss of *Egln1*, thereby stabilising HIF signalling in all cell types but restricted to epicardium in explants for ex vivo analysis (*Figure 3a–d*; *Figure 3—figure supplement 1*). Upon tamoxifen treatment, *Rosa26$^{+/CreERT2}$; Egln1$^{fl/fl}$* derived explants (KO) exhibited a significant induction of HIF-1α and HIF-2α expression as compared to controls (CTR; *Figure 3—figure supplement 1a–c*; HIF-1α; CTR: 0.26±0.13; KO: 0.86±0.09; p=0.0061; HIF-2α; CTR: 0.37±0.01;

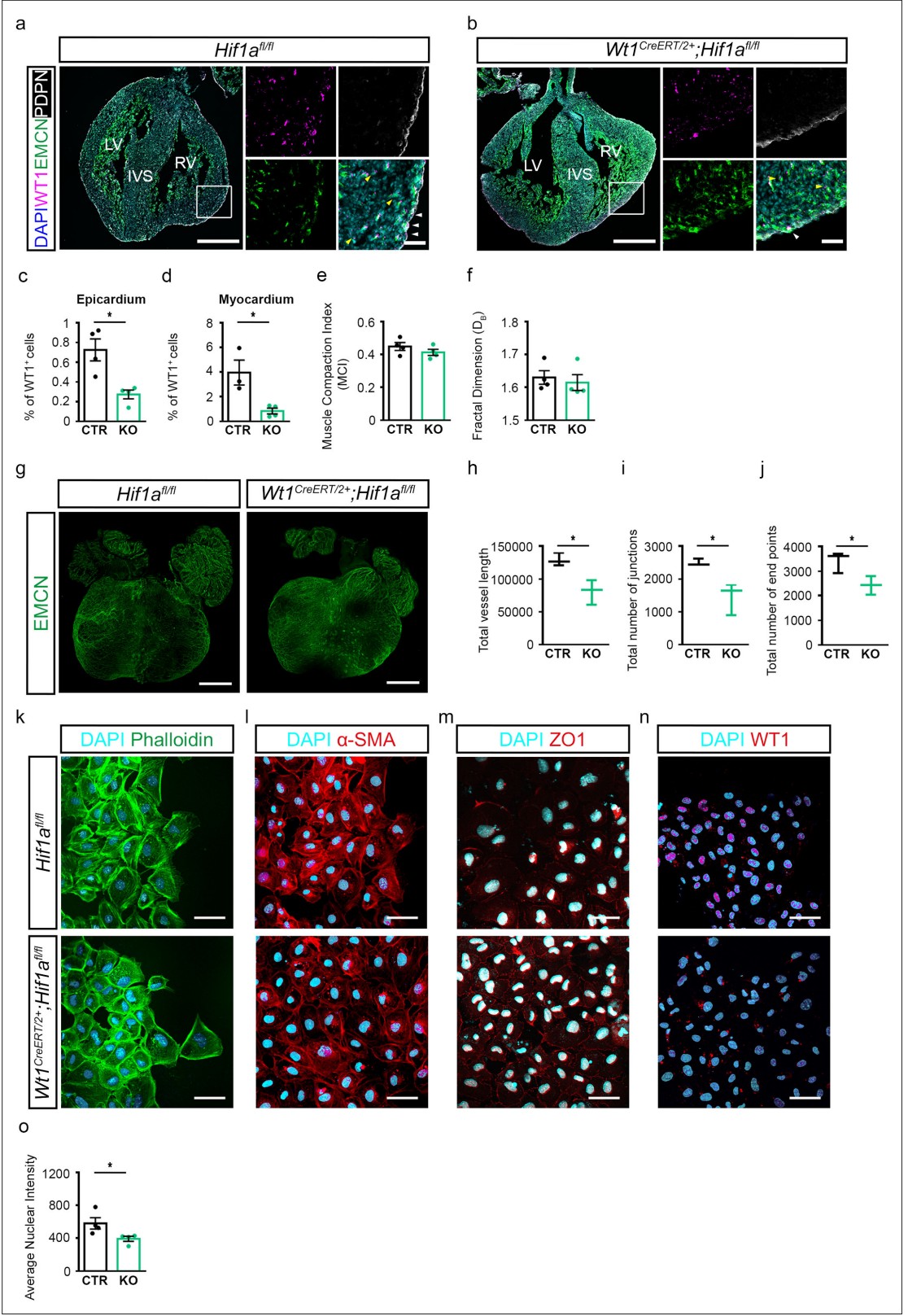

**Figure 2.** Epicardial loss of *Hif1a* leads to reduced WT1 expression and impaired EMT. (**a, b**) Representative images of immunostaining for EMCN (green), WT1 (magenta), PDPN (white), and DAPI (blue), on coronal sections of hearts from (**a**) *Hif1a^fl/fl* (CTR) or (**b**) *Wt1^CreERT2/+;Hif1a^fl/fl* (KO) embryos injected with tamoxifen at E9.5 and E10.5 and harvested at E16.5. Images on the right of each panel represent high magnifications of boxed regions. White arrowheads indicate WT1-expressing cells in the epicardium. Yellow arrowheads indicate WT1-expressing cells in the myocardial compartment.

*Figure 2 continued on next page*

*Figure 2 continued*

Quantification of WT1+ cells in (**c**) the epicardium (CTR, n=4; KO, n=4) or (**d**) the myocardium (CTR, n=4; KO, n=4). (**e**) Muscle compaction index and (**f**) fractal dimension analysis to assess the complexity of myocardial trabeculae. (CTR, n=4; KO, n=4). (**g**) Whole-mount immunostaining for EMCN (green) to visualise the coronary vasculature of CTR and KO embryos at E16. 5. (**h–j**) Quantification of coronary vasculature as (**h**) total vessel length, (**i**) number of junctions, and (**j**) end points carried out using AngioTool. n=3 hearts/group. Representative images of immunostaining for (**k**) Phalloidin (green), (**l**) α smooth muscle actin (α-SMA, red), (**m**) zonula occludens-1 (ZO-1, red), (**n**) WT1 (red) and DAPI nuclear stain (cyan) on epicardial explants derived from *Hif1a*$^{fl/fl}$ (CTR) and *Wt1*$^{CreERT2/+}$;*Hif1a*$^{fl/fl}$(KO) hearts harvested at E11.5. (**o**) Quantification of nuclear intensity of WT1 signal on epicardial explants (CTR, n=4; KO, n=4). IVS = interventricular septum, LV = left ventricle, RV = right ventricle. Whole heart scale bars, 500 µm; high-magnification scale bars, 50 µm. Data presented as median, inter-quartile range (IQR) and upper and lower limits. Error bars represent mean ± s.e.m. *n* numbers refer to individual mice. Two-tailed, unpaired Student t-tests were used for statistical analysis. *p<0.05.

The online version of this article includes the following figure supplement(s) for figure 2:

**Figure supplement 1.** Epicardial deletion of HIF-1α does not alter HIF-2α expression nor proliferation of WT1$^+$ cells.

**Figure supplement 2.** Tamoxifen treatment of *Wt1*$^{CreERT2/+}$;*Hif1a*$^{fl/fl}$ -derived epicardial explants reduces HIF-1α expression without affecting HIF-2α.

**Figure supplement 3.** HIF-1β subunit directly binds to the WT*1* locus.

KO: 1.12±0.15; p=0.0027). Subsequent visualisation of stress fibres via α-SMA staining revealed augmented mesenchymal morphology in the mutant-derived explants (*Figure 3a*). Consistent with this, WT1 nuclear fluorescence was significantly increased in KO explants as compared to controls (CTR; *Figure 3b and c*; CTR: 158.7±48.34; KO: 330.5±27.26; p=0.0364). In addition, expression levels of the EMT transcription factor snail family transcriptional repressor 2 (*Snai2*) gene were increased in *Egln1*-KO explants relative to CTR (*Figure 3d*; CTR: 1±0.025; KO: 1.166±0.036; p=0.0464).

To complement our genetic studies, we undertook a more therapeutically relevant approach by using a pharmacological PHD inhibitor to stabilise HIF signalling. Epicardial explants were generated from C57BL/6 wildtype embryos and treated with DMSO control (CTR) or Molidustat (BAY 85–3934, Mol) a specific PHD inhibitor, used in clinical trials for renal anaemia (*Macdougall et al., 2019*; *Figure 3e–j*). Immunostaining for HIF-1α and HIF-2α confirmed the stabilisation of HIF signalling upon PHD inhibitor treatment (*Figure 3—figure supplement 1d–g*). Staining with phalloidin (*Figure 3e*) and α-SMA (*Figure 3f*) showed an enhanced mesenchymal morphology with increased stress fibres in inhibitor-treated explants, suggestive of EMT induction. Furthermore, immunostaining demonstrated a significant increase in WT1 expression in treated explants as compared to control (*Figure 3g and h*; CTR: 1039±134.9; Mol: 2500±503.7; *P*=0.03). Likewise, the PHD inhibitor enhanced the motility of cultured mouse embryonic ventricular epicardial cells (MEC.1 cell line) (*Li et al., 2011*) compared to control-treated cells (*Figure 3i and j*; CTR: 436869±9359; Mol: 372482±20880; p=0.0477), as assessed by scratch assays. These findings collectively suggest that genetic perturbation and/or pharmacological inhibition of PHDs stabilise HIF signalling in the epicardium ex vivo, leading to increased WT1 expression and stimulation of epicardial EMT.

## Reduced HIF signalling is associated with epicardial quiescence in the neonatal heart

To gain unbiased insight into hypoxia and HIF-related molecular pathways which are differentially regulated in intact P1 versus P7 mouse hearts, we performed single cell RNA-seq analysis using the 10x Genomics Chromium platform and next generation sequencing. Cell clusters were visualised through uniform manifold approximation and projection (UMAP; *Figure 4a*). An epicardial cell cluster (Epi) was identified based on previously described specific epicardial gene expression signatures (*Bochmann et al., 2010*; *Figure 4b*). As expected, a cardiomyocyte cell cluster (CM) was underrepresented due to both low survival and size incompatibility with the FACS cell sorting (*Cui and Olson, 2020*). Gene Ontology (GO) analysis for biological processes showed an enrichment of hypoxia-related pathways in P1 compared to P7 hearts (*Figure 4—figure supplement 1a*), particularly in the epicardial cell population (*Figure 4c*). In contrast, hypoxia-related pathways were enriched in P7 hearts compared to P1 hearts in the fibroblast cluster (*Figure 4—figure supplement 1b–d*), suggesting a role for hypoxia post-injury, specifically in the fibrotic response leading to scarring. Accordingly, expression of well-known HIF-induced genes such as *Vegfa* and pyruvate dehydrogenase kinase 3 (*Pdk3*) was increased in P1-derived epicardial cells (*Figure 4d*). Conversely, the expression of *Egln1*, encoding for the main suppressor of HIF signalling, was enriched in P7 cells (*Figure 4e*). These findings were further confirmed by a time-course analysis of mRNA levels using ventricle lysates and immunostaining which

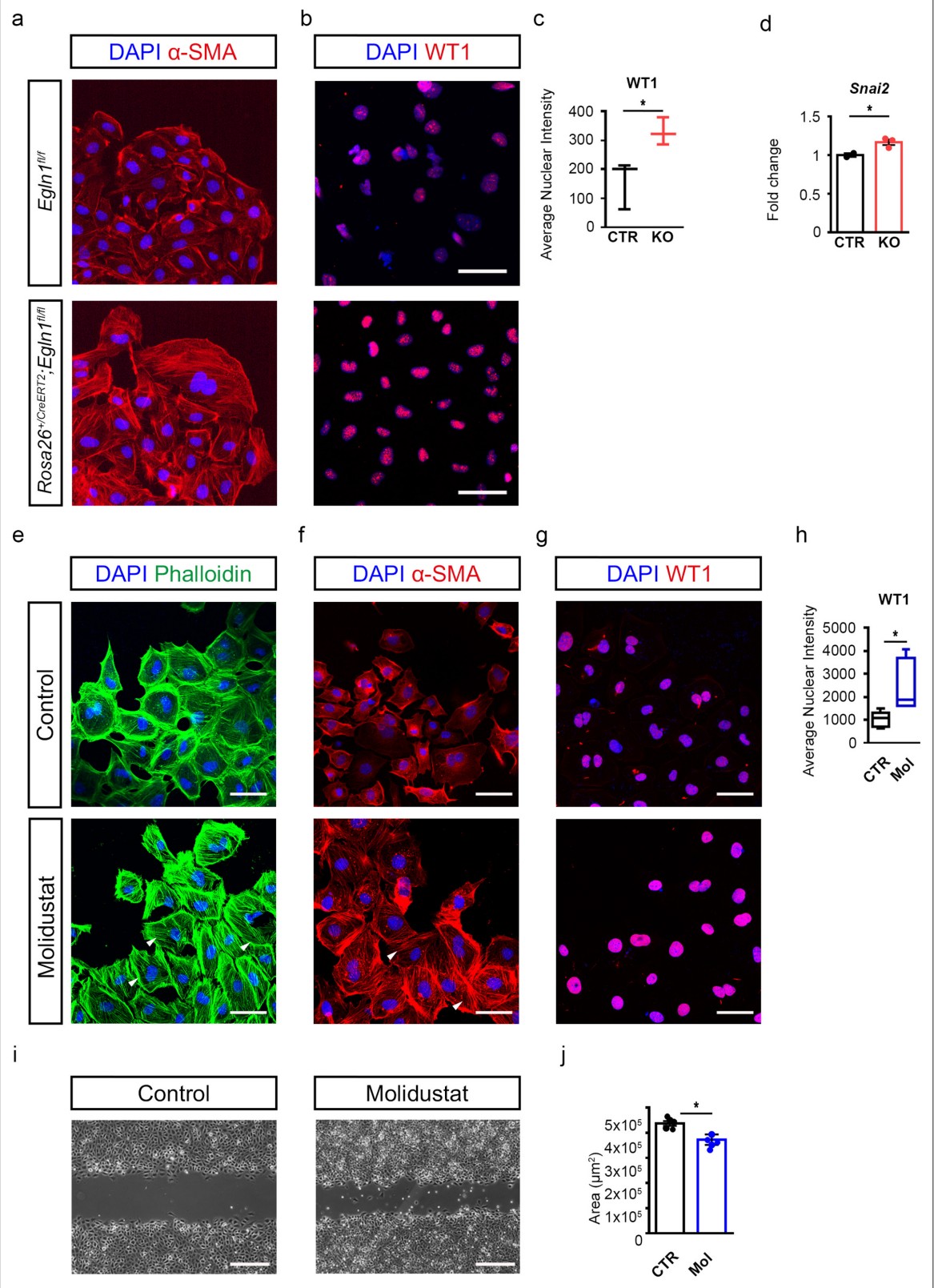

**Figure 3.** Stabilisation of HIF signalling promotes Wt1 expression and enhances epicardial EMT. (**a, b**) Representative images of immunostaining for α smooth muscle actin (α-SMA, red), WT1 (red), and DAPI nuclear stain (blue) and (**c**) quantification of nuclear intensity of WT1 signal on epicardial explants derived from *Egln1^fl/fl^* (CTR) and *Rosa26^+/CreERT2^; Egln1^fl/fl^* (KO) embryos at E11.5. n=5 hearts/group. (**d**) *Snai2* expression analysis by qRT-PCR using RNA isolated from epicardial explants derived from CTR (n=2) and KO (n=3) hearts. (**e–g**) Representative images of immunostaining for Phalloidin

*Figure 3 continued on next page*

*Figure 3 continued*

(green), α-SMA (red), WT1 (red) and DAPI nuclear stain (blue), and (**h**) quantification of nuclear intensity of WT1 signal on epicardial explants derived from wild type hearts harvested at E11.5 and treated with DMSO (control, CTR, n=6) or Molidustat (Mol, n=6). Scale bars, 50 μm. Arrowheads indicate stress fibres. *n* numbers refer to individual mice. (**i**) Representative images and (**j**) quantification of cell migration assessed by wound healing/scratch assay of MEC.1 cells treated with DMSO (control, CTR, n=5), or Molidustat (Mol, n=5). *n* numbers refer to technical replicates. Scale bars, 0.5 μm. Data presented as median, inter-quartile range (IQR) and upper and lower limits or mean ± *s.e.m.* Two-tailed, unpaired Student t-tests were used for statistical analysis. *p<0.05.

The online version of this article includes the following figure supplement(s) for figure 3:

**Figure supplement 1.** Genetic and pharmacological stabilisation of HIF signalling induces both HIF-1α and HIF-2α expression in epicardial explants.

revealed a marked upregulation of *Egln1* cardiac levels from P7 to adulthood (***Figure 4f***, ***Figure 4— figure supplement 2a, b***). Notably, this increase in *Egln1* levels coincided with a reduction in WT1 expression in P7 versus P1 hearts (***Figure 4g***).

## Activation of HIF signalling improves the response to injury in non-regenerative P7 hearts

To establish a causative role in the regulation of epicardial WT1 activity, we extended HIF signalling beyond P7 using a genetic gain-of-function approach. Neonatal *Wt1^{CreERT2/+}*; *Egln1^{fl/fl}* pups received a single dose of tamoxifen at P2 to induce Cre-mediated loss of PHD2. MI was then induced by permanent ligation of the proximal left anterior descending (LAD) coronary artery in P7 mice (***Figure 5a***), a time-point when the heart's regenerative capacity is lost. Hearts were collected at 21 dpi. Immunostaining analysis of the endothelial marker CD31 and α-SMA did not reveal a significant difference in the number of vessels between *Egln1^{fl/fl}* (CTR) and *Wt1^{CreERT2}*; *Egln1^{fl/fl}* (KO) hearts (***Figure 5b and c***, CTR 0.0030±0.0004, KO 0.0034±0.0002, p=0.45), suggesting that activation of HIF signalling in WT1-expressing cells does not affect the neovascularisation response. To investigate whether epicardial activation of HIF signalling improved heart function following MI, we performed cine-MRI analysis at 3 weeks post-MI. We observed a substantial increase in the ejection fraction (EF) and decrease in the end-diastolic volume (EDV) in KO animals compared to controls (***Figure 5d–f***; EF: CTR 43.21±6.25, KO 64.30±6.82, p<0.05. EDV; CTR 54.19±3.50, KO 33.72±2.67; p<0.05). Furthermore, histological analysis by Masson's trichrome staining revealed a significant reduction of the fibrotic scar in KO animals as compared to controls (***Figure 5g and h***, CTR 15.44±1.088, KO 10.34±1.374, p=0.027) without changes in the size of cardiomyocytes, as assessed by wheat germ agglutinin staining (WGA; ***Figure 5i and j***, CTR 167.6±3.27, KO 179.3±8.68, p=0.32). Taken together, these findings suggest that activated hypoxia signalling in the epicardium contributes to heart regeneration by reducing scar size and improving heart function.

To further investigate our hypothesis that prolonged maintenance of HIF signalling has a beneficial effect on the epicardium and response to MI in the postnatal heart, we explored a complementary pharmacological approach recapitulating our gain of HIF-function studies using epicardial explants. We induced MI by LAD ligation in P7 mice and administered Molidustat or DMSO control (vehicle) by intraperitoneal injection, immediately after surgery and one week later, to stabilise HIF signalling in the postnatal heart (***Figure 6a***). Treatment with PHD inhibitor effectively stabilised HIF-1α and HIF-2α, as observed by immunostaining (***Figure 6—figure supplement 1a, b***) in hearts collected at 9 days post-MI (9dpi). Moreover, this pharmacological approach revealed a significant increase in epicardial WT1 expression in both the infarct and remote zones (***Figure 6b***) in Molidustat-treated hearts (MI T) compared to controls (CTR) (***Figure 6c and d***, infarct zone: MI CTR 0.34±0.08, MI T 0.50±0.06, remote zone: CTR 0.075±0.02, MI T 0.25±0.05, p<0.05). Similarly, an increase in WT1 expression in the myocardium was observed in the PHD-inhibitor treated group and was more pronounced proximal to the infarct region (***Figure 6e and f***, infarct zone: CTR 5.02±1.34, MI T 7.09±1.86, p=<0.05; remote zone: CTR 13.88±4.96, MI T 18.61±6.18; p<0.05). We observed an extensive co-localisation of WT1 and EMCN within the myocardial compartment at 9 dpi (***Figure 6—figure supplement 1c***), supporting an endothelial identity for these cells. Furthermore, an assessment of the proliferation of WT1+ cells and cardiomyocytes revealed no significant differences between control and Molidustat-treated animals (***Figure 6—figure supplement 1d, e, g***).

To analyse the long-term histological effects of HIF-stabilisation after MI, we collected hearts at 21 dpi. Co-immunostaining and quantification of CD31 and α-SMA revealed no difference in the number

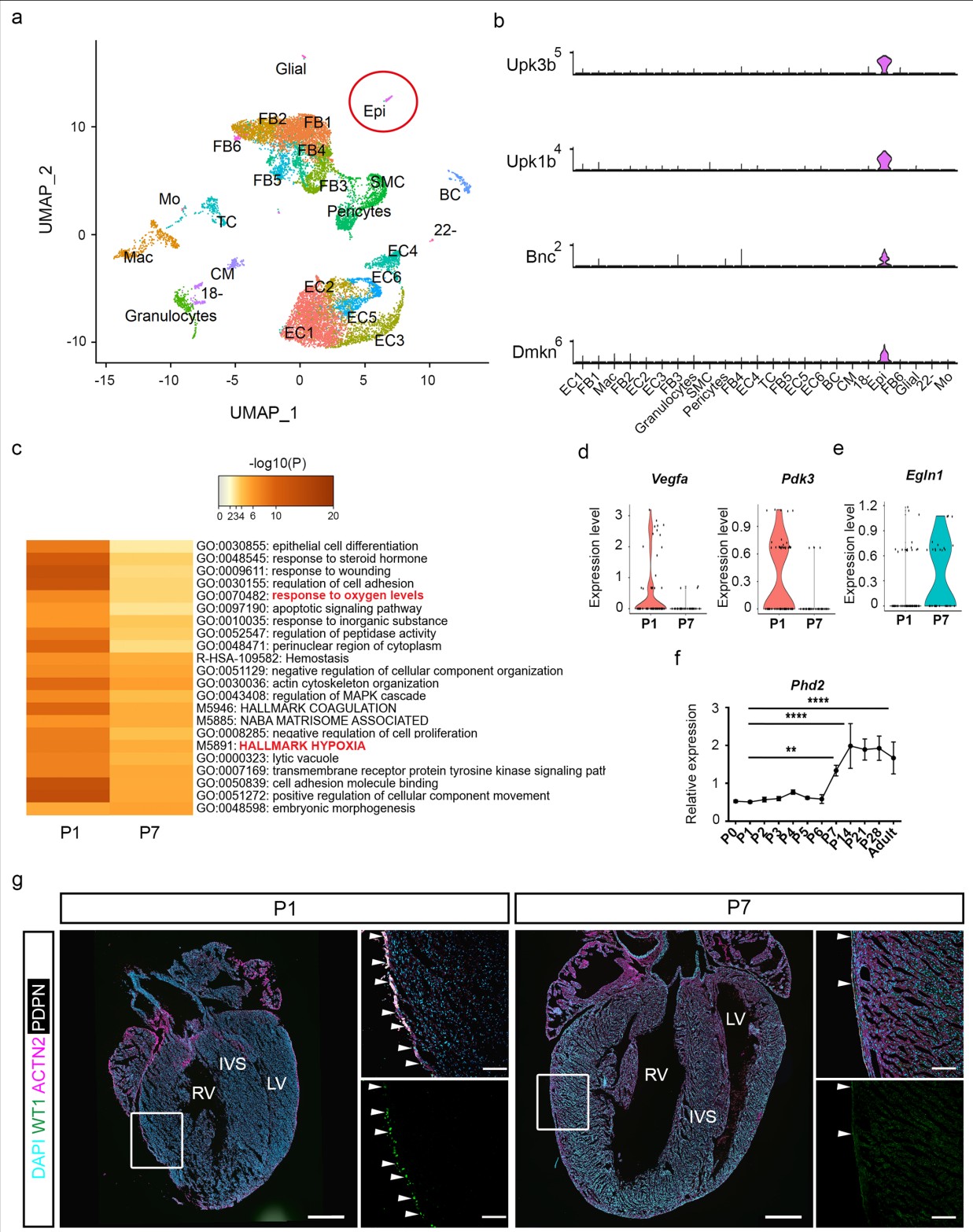

**Figure 4.** HIF signalling is downregulated in between P1 and P7 in neonatal mice. (**a**) UMAP representation of different cell populations in the intact neonatal heart at postnatal day (P) 1 and P7. (**b**) Stacked violin plots showing expression of canonical Epi-enriched genes *Upk3b*, *Upk1b*, *Bnc*, *Dmkn*. (**c**) Heatmap showing biological processes enriched in the Epi cluster at P1 versus P7. Violin plots showing expression of HIF target genes (**d**) *Vegfa* and *Pdk3*, and (**e**) *Egln1* at P1 versus P7. (**f**) Real time RT-PCR analysis of *Egln1*, n=4 hearts per group. (**g**) Representative images of immunostaining for WT1 (green), ACTN2 (magenta), PDPN (white), and DAPI nuclear stain (cyan) at P1 and P7. Images to the right are magnified views of boxed regions shown

*Figure 4 continued on next page*

*Figure 4 continued*

in whole heart images. Arrowheads indicate expression of WT1 in the epicardium. IVS = interventricular septum, LV = left ventricle, RV = right ventricle. Whole heart scale bars, 200 μm; high-magnification scale bars, 100 μm. n=4 hearts per stage. Data are presented as mean ± s.e.m. *n* numbers refer to individual mice. One-way ANOVA with Tukey's post-hoc tests was used for statistical analysis. **p<0.01; ****p<0.0001. Epi, epicardium.

The online version of this article includes the following figure supplement(s) for figure 4:

**Figure supplement 1.** Differential activity of HIF signalling in the neonatal mouse.

**Figure supplement 2.** PHD2 expression increases in the neonatal heart from P1 to P7.

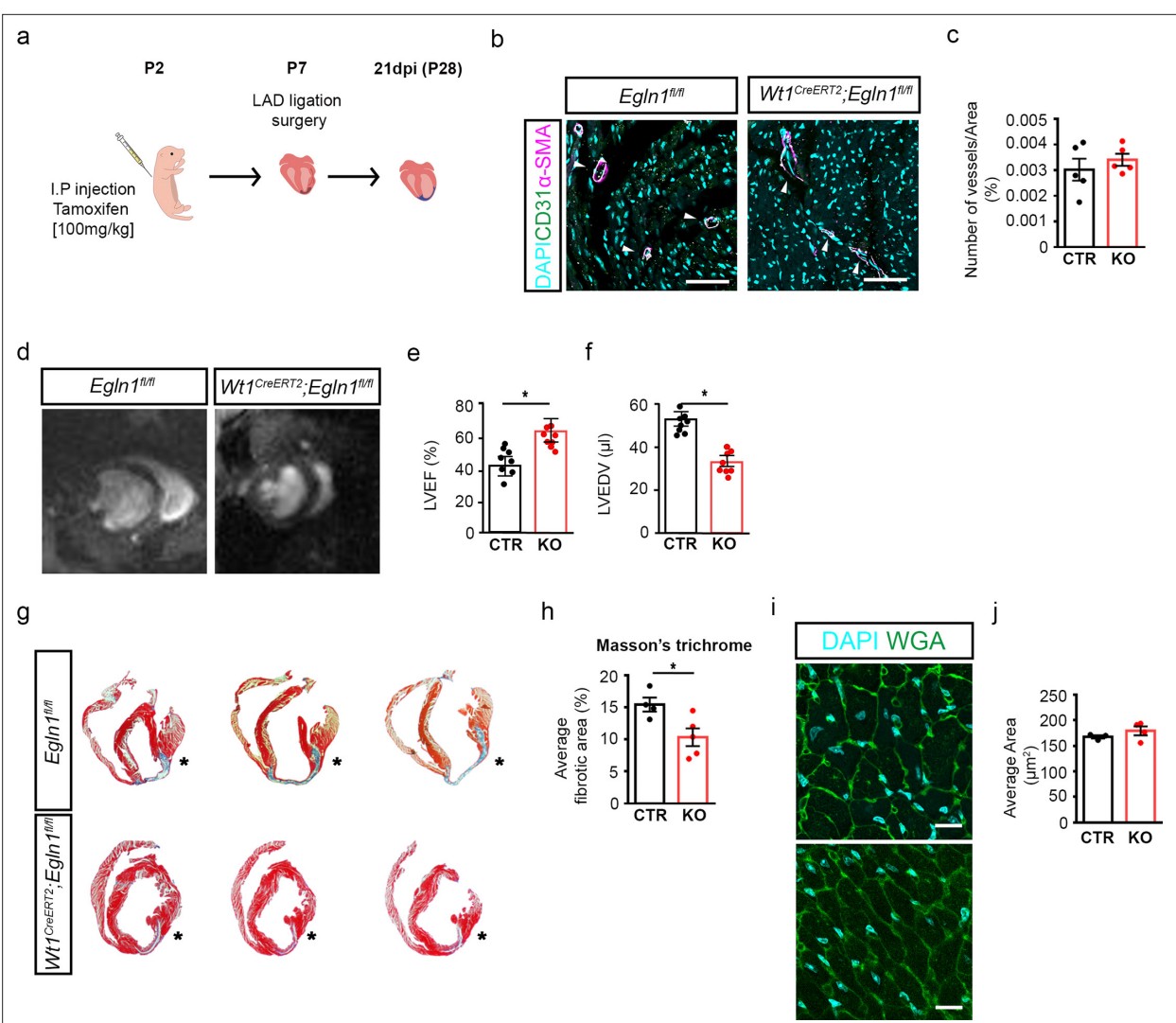

**Figure 5.** Loss of *Egln1* (PHD2) in WT1 expressing cells improves heart regeneration post-MI. (**a**) Schematic of experimental design. (**b**) Representative images of immunostaining for α-smooth muscle actin (α-SMA, magenta), CD31 (green), and DAPI nuclear stain (cyan) on sections of hearts from *Egln1^{fl/fl}* (CTR, n=5) and *Wt1^{CreERT2}; Egln1^{fl/fl}* (KO, n=5) mice and (**c**) quantification of number of vessels at 21 days post-injury (dpi). Arrowheads indicate vessels (determined by co-expression of CD31 and α-SMA). Scale bars, 100 μm. (**d**) Representative mid-ventricular short-axis MRI frames for *Egln1^{fl/fl}* (CTR) and *Wt1^{CreERT2}; Egln1^{fl/fl}* (KO) mice hearts. (**e, f**) MRI analyses of infarcted hearts at 21 dpi showing increased EF (**e**) and reduced EDV (**f**) in *Wt1^{CreERT2}; Egln1^{fl/fl}* (KO, n=8) animals compared with controls (CTR, n=8). (**g**) Representative images and (**h**) quantification of Masson's Trichrome stained transverse serial sections to assess cardiac fibrosis (blue) at 21 dpi. * denotes suture placement. (**i**) Representative images of immunostaining for wheat-germ agglutinin (WGA, green) and DAPI nuclear stain (cyan) on hearts from control and Molidustat treated mice 21dpi for: Scale bars, 100μm. (**g**) Quantification of cardiomyocyte cell size, as assessed by WGA staining on sections of hearts from CTR (n=4) and KO (n=4). Left ventricular regions were measured. Error bars represent mean ± *s.e.m. n* numbers refer to individual mice. Two-tailed, unpaired Student t-tests were used for statistical analysis. *p<0.05.

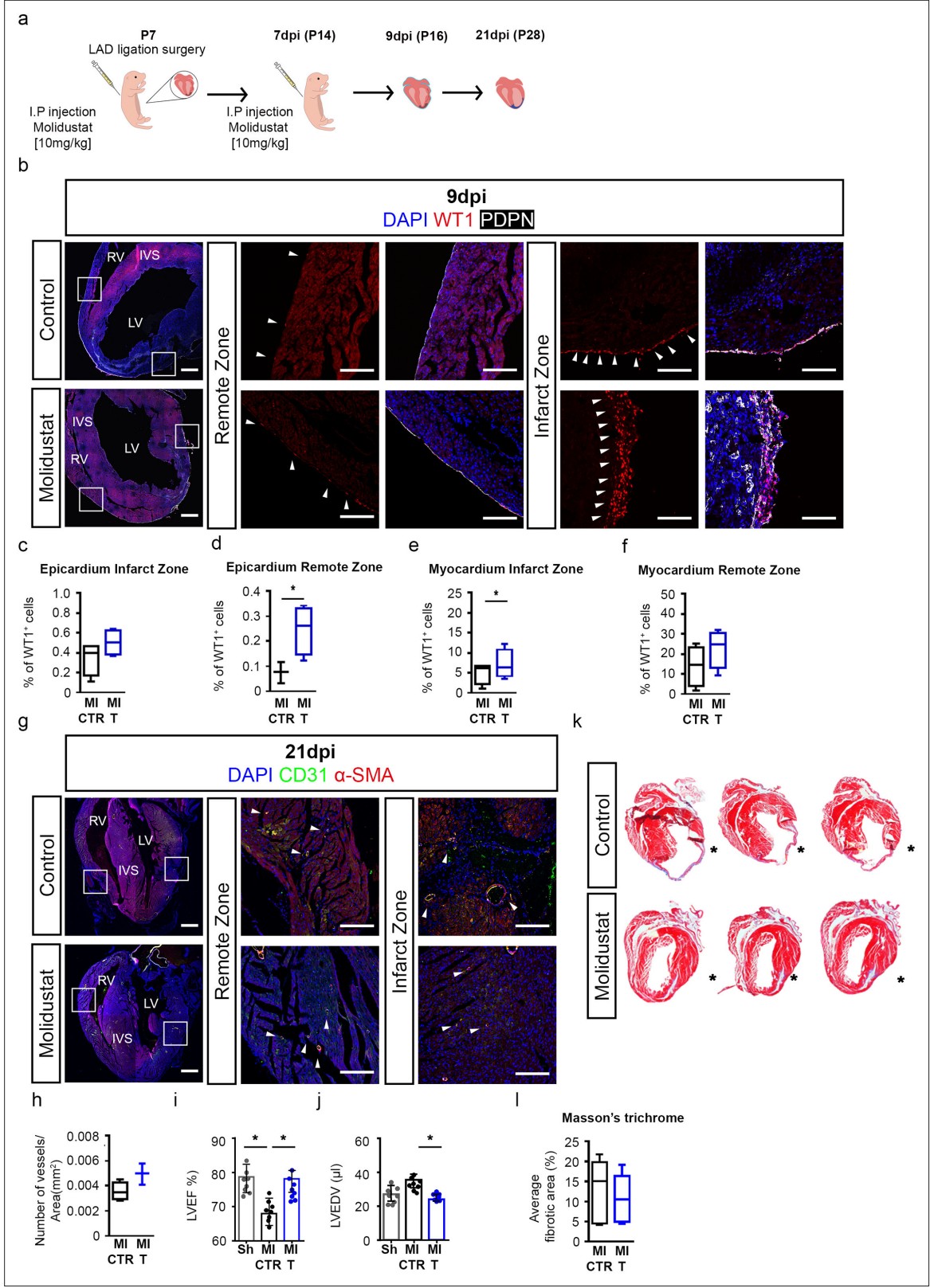

**Figure 6.** Pharmacological treatment with PHD inhibitors induces WT1 expression and improves heart function. (**a**) Schematic of experimental design. (**b**) Representative images of immunostaining for WT1 (red), PDPN (white), and DAPI nuclear stain (blue) on sections of hearts from animals treated with either saline (MI CTR) or Molidustat (MI T) and (**c–f**) quantification of number of WT1 positive cells at 9 days post-injury (dpi). Images on the right represent high magnification views of boxed regions shown in whole heart images. Arrowheads indicate expression of WT1 in the epicardium

*Figure 6 continued on next page*

*Figure 6 continued*

(determined by co-expression with PDPN). n=4 animals per group. (**g**) Representative images of immunostaining for CD31 (green), α-SMA (red), and DAPI (blue) on sections of hearts from animals treated with either saline (MI CTR, n=4) or Molidustat (MI T, n=4) and (**h**) quantification of number of vessels at 21dpi. Arrowheads indicate vessel (determined by co-expression of CD31 and α-SMA). Whole heart scale bars, 200 μm; high-magnification scale bars, 100 μm. Left ventricle (**i**) Ejection fraction (LVEF) and (**j**) end diastolic volume (LVEDV) evaluated by MRI analyses of non-infarcted (Sham, Sh; n=8), infarcted (MI CTR; n=8) controls and treated (MI T; n=8) hearts at 21dpi. (**k**) Representative images and (**l**) quantification of Masson's Trichrome stained transverse sections to assess cardiac fibrosis (blue) at 21 dpi. * denotes suture placement. Scale bars, 1 mm. IVS = interventricular septum, LV = left ventricle, RV = right ventricle. Scale bars 200 μm, high-magnification scale bars, 50 μm. Data presented as median, IQR and upper and lower limits and as mean ± s.e.m. *n* numbers refer to individual mice. Two-tailed, unpaired Student *t*-tests were used for statistical analysis between two groups, one-way ANOVA with Dunnett's post-hoc test was used for multiple comparisons in i and j. *p<0.05.

The online version of this article includes the following figure supplement(s) for figure 6:

**Figure supplement 1.** Effects of PHD inhibitor Molidustat administration to infarcted neonatal hearts at P7 and P14 on HIF signalling, cell proliferation, and hypertrophy.

of vessels between controls and the treatment group (*Figure 6g and h*). Cine-MRI analysis showed a marked reduction in left ventricular systolic function following MI in the vehicle-treated group (MI CTR) as compared to uninjured hearts (Sh) and, more importantly, a significant improvement in systolic function in the PHD inhibitor treated groups (MI T). EF and EDV were significantly improved in the drug-treated groups (*Figure 6i and j*; EF: Sh 78.10±3.22, MI CTR 67.70±4.31, MI T 77.61±1.76, p<0.05; EDV: Sh 27.53±3.10, MI CTR 36.11±2.49 MI T 26.44±0.93; p<0.05). Histological analysis by Masson's trichrome staining revealed the presence of a fibrotic scar in drug-treated animals, albeit this was reduced relative to controls (*Figure 6k and l*) and the at-risk myocardium was comparatively thicker with minimal dilatation or thinning of the left ventricle. The thickened myocardium was not due to an effect on cardiomyocyte hypertrophy, as determined by WGA staining (*Figure 6—figure supplement 1f–h*) nor an effect on cardiomyocyte proliferation (*Figure 6—figure supplement 1e, g*) suggesting potential cardio-protection and improved preservation of survived heart muscle. More generally, these findings also indicate that improved remodelling and function following drug treatment are possible against a background of fibrosis.

## Discussion

In this study, we characterised the levels of hypoxia in the heart during mid-to-late gestation when the epicardium is fully formed. At E12.5, hypoxia was mainly observed in areas of compact myocardium, most significantly in the atrioventricular (AV) groove, close to where the coronary plexus first develops (*Chen et al., 2014*). By E14.5, hypoxic regions localised to the interventricular septum (IVS), which is perfused later than the muscle of the free wall (*Tian et al., 2013*). Finally, by E16.5 and E18.5, when the heart is adequately perfused by the coronary vessels, hypoxia was primarily restricted to the epicardium. We further analysed the expression and distribution analysis of HIF-1 isoforms, finding a predominance of the HIF-1α isoform in WT1+ cells, whereas HIF-2α was more localised in the myocardium. Given hypoxia's known role in promoting EMT (*Scully et al., 2016*), we confirmed a functional role for HIF-1α in promoting morphological changes associated with EMT by direct regulation of *Wt1* expression (*Figures 2 and 3*, *Figure 2—figure supplement 3*). Moreover, epicardial-specific deletion of *Hif1a* significantly reduced the number of WT1-expressing cells in both epicardial and myocardial compartments, along with impaired coronary vessel development in E16.5 embryos. Collectively, this suggests that HIF-1α induces epicardial EMT and plays an essential role in coronary vessel formation during development. While Cre-mediated deletion of HIF-1α, induced at E9.5 and E10.5, predominantly targeted epicardial cells, it is worth noting that expression of WT1 in the coronary endothelium has been reported as early as E11.5 (*Lupu et al., 2020*), although the biological significance of endothelial WT1 remains elusive. Thus, aberrant HIF signalling in WT1 +endothelial cells cannot be completely excluded as contributing to the observed defects on the expansion of the coronary vasculature. However, stabilisation of HIF signalling in epicardial explant cultures, under physiological oxygen levels, either via genetic ablation of *Egln1* or chemical inhibition of PHD enzymes with Molidustat, resulted in enhanced EMT and WT1 expression, further confirming a role for HIF signalling in epicardial activation.

To gain insight into the post-natal molecular pathways regulating the epicardium, we performed single cell RNA sequencing analysis comparing P1 and P7 stages, focusing on the specific gene expression signature of the epicardial cell cluster. GO terms analysis showed an enrichment of hypoxia-related pathways in P1 hearts, consistent with the expression of well-known HIF target genes. Importantly, *Egln1* expression was enriched in P7 hearts, potentially contributing to epicardial quiescence during the first week of life, concurrent with a decrease in HIF signalling. Complete regeneration following MI requires both the replacement of lost cardiomyocytes and the formation of new blood vessels. Given the role of the epicardium during development in promoting coronary vessel formation and myocardial growth, the observed epicardial quiescence after birth likely contributes to the loss of regenerative capacity. Genetic and pharmacological stabilisation of HIF signalling beyond P7 proved to be effective in extending the regenerative window after LAD surgery, resulting in reduced pathological remodelling and preserved cardiac function against a background of fibrotic repair. Interestingly, genetic stabilisation of HIF signalling specifically in WT1-expressing cells significantly reduced scar size post-injury (*Figure 5*). Reduced fibrosis was also observed in PHD-inhibitor treated P7 mice (*Figure 6*). This is important since necrotic cardiomyocytes can trigger the activation of fibroblasts and promote fibrosis (*Prabhu and Frangogiannis, 2016*). The epicardium is a key source of mitogens in the embryonic heart (*Simões and Riley, 2018*) and provides paracrine signals targeting cardiomyocytes during regeneration (*Zhou and Pu, 2011*; *Wills et al., 2008*). In our study, we did not observe any change in either cardiomyocyte proliferation or hypertrophy, strengthening the hypothesis of an epicardium-mediated role in cardiomyocyte protection from apoptosis via paracrine signalling (*Zhou et al., 2011*).

It is noteworthy that despite persistent scarring (albeit reduced) following either genetic or pharmacological perturbation of PHDs, significant functional improvements were observed, in contrast to the prevailing view that fibrotic repair after MI is a barrier to effective tissue regeneration (*Liang et al., 2019*; *Simões et al., 2020*; *Koth et al., 2020*). Previous studies have suggested the involvement of HIF-1α in myocardial remodelling after injury. Mice with constitutive overexpression of HIF-1α in the myocardium showed enhanced angiogenesis, attenuation of infarct size, and improved cardiac performance after MI (*Kido et al., 2005*). Furthermore, cardiac-specific PHD2 inactivation and consequent HIF activation play a causal role in the pathogenesis of ischaemic cardiomyopathy (*Moslehi et al., 2010*). In our study, we focused on the stabilisation of HIF signalling as a therapeutic approach to improve outcomes after heart injury. The mean half-life of Molidustat ranges from 4 to 10 hours (*Böttcher et al., 2018*), ensuring prolonged stabilisation of HIF signalling to enable therapeutic benefit. Molidustat is an orally administered small molecule used for the treatment of anaemia in patients with non-dialysis-dependent chronic kidney disease (CKD) (*Macdougall et al., 2019*) and could potentially be repurposed to treat ischaemic heart disease.

In summary, we show that the epicardium is hypoxic at later stages of development, with HIF-1α playing a crucial role in epicardial activation and EMT, necessary for supporting coronary vessel development. The epicardium becomes quiescent after birth, coinciding with decreased HIF-signalling. However, maintaining HIF signalling after birth can extend epicardial activation, leading to a significant improvement in cardiac remodelling and function after injury, thus representing a potential novel therapeutic target to improve cardiac regeneration following MI.

## Materials and methods
### Mouse Lines

All mice were housed in individually ventilated cages (IVCs) and ventilated racks at 22 °C and 55% humidity, under controlled environmental conditions according to the United Kingdom Home Office. The following mouse lines were used: Rosa$^{26+/CreERT2}$;*Hif1a*$^{fl/fl}$; Rosa26$^{+/CreERT2}$; *Egln1*$^{fl/fl}$. The *Wt1*$^{CreERT2/+}$;*Hif1a*$^{fl/fl}$ and *Wt1*$^{CreERT2/+}$; *Egln1*$^{fl/lf}$ mouse line was generated by crossing *Wt1*$^{CreERT2/+}$ (*Fan et al., 2019*) with *Hif1a*$^{fl/fl}$ animals or *Egln1*$^{fl/lf}$ for two generations. Genetically modified mouse lines used were kept in a pure C57BL/6 background. Both males and females were used in the study. For timed-mating experiments, 8- to 12-week-old mice were set up overnight and females checked for vaginal plugs the following morning; the date of a vaginal plug was set as embryonic day (E) 0.5. For tamoxifen-dependent gene activation, 2 doses of 40 mg/kg of body weight of tamoxifen (Sigma) were administered to pregnant dams by oral gavage, at embryonic stages E9.5 and E10.5. For neonate studies,

pups were injected intraperitoneally (i.p.) with a single 10 µl dose of 20 mg/ml tamoxifen, at postnatal day (P)2, using a 25-gauge needle. 5-ethynyl-2'-deoxyuridine (EdU, Thermofisher) was intraperitoneally injected at a dose of 50 mg/kg at the day of surgery (P7) and every other day. Hearts were collected at 9 dpi and the assay was carried out according to manufacturer's protocol.

## Hypoxyprobe

For hypoxia studies, pregnant females were injected i.p. with 1.5 mg Hypoxyprobe (pimonidazole hydrochloride, Hypoxyprobe-1 Inc). After 2 hr, hearts from embryos were harvested and fixed. Pimonidazole is a 2-nitroimidazole that is reductively activated specifically in hypoxic cells and forms stable adducts with thiol groups in proteins, peptides, and amino acids at oxygen levels below 1.3%.

## Cell lines

The mouse immortalised embryonic epicardial cell line MEC.1 (*Zhou et al., 2008*) was purchased from Merck/Millipore (SCC187) and cultured according to manufacturer' recommendations.

## Myocardial infarction

All procedures were carried out following local ethical approval (AWERB) at the University of Oxford and under the regulation of a United Kingdom Home Office project licences (references PPL30/2987, PPL30/3155, PDDE89C84 and PP3194787) in full compliance with the Animals (Scientific Procedures) Act 1986 (A(SP)A, revised 2012). Animals were euthanised according to the UK Home Office designated schedule 1 method of cervical dislocation. MI was induced by permanent ligation of the left anterior descending (LAD) coronary artery in wild type mice at postnatal day (P) 7, as previously described (*De Villiers and Riley, 2021*). All surgery was performed under isoflurane anaesthesia, and every effort was made to minimise suffering. Neonates cannot be sexed during the first week of life, so mixed male and female cohorts were used throughout the protocol. Mice were anaesthetised with 4% isoflurane for 1 min and then transferred to an ice box for up to 6 min. Mice were recovered under 0.5% isoflurane. Animals received intraperitoneal injections of BAY 85–3934 (Molidustat, Selleck) (10 mg/kg) or vehicle (DMSO), upon recovery (day 0) and 7 days later. Mice were sacrificed by cervical dislocation for tissue collection at 4-, 9- and 21 days following ligation. Mice with excessive or minimal infarct sizes at early time points were excluded from further histological or functional assessment (see under Cine MRI). Investigators were blinded to mouse genotype and treatments prior to sample analyses.

## Epicardial explants

Sterile 12-well plates (Fisher Scientific), containing sterile 13 mm diameter coverslips were coated with 0.1% gelatin (Millipore) and allowed to stand for 20 min. The gelatin was then replaced by Dulbecco's Modified Eagle Medium (DMEM) (Sigma) containing 10% FBS (Sigma), 1% Penicillin/Streptomycin (Sigma). Hearts were isolated from E11.5 embryos and the outflow tract and atria were removed. Each ventricle was then cut in half and placed epicardial side down on the coverslip. Explants were kept at 37 °C/5% $CO_2$. 24 hr later, 1 µM of tamoxifen was added to explants. After 48 hr of culture, the explant tissue was carefully peeled off, leaving only the epicardial sheet remaining on the coverslip. For explants prepared from C57BL/6 embryos, 50 µM of BAY 85–3934 (Molidustat, Selleck) or vehicle (DMSO) was added. After a total of 72 hr in culture, the media was removed, coverslips were washed briefly in ice-cold PBS and then fixed in 4% PFA at room temperature for 15 min. The MEC.1 cell line identity was authenticated by qPCR and immunostaining for known epicardial markers and tested negative for mycoplasma contamination. Coverslips were subsequently washed in PBS before proceeding with immunostaining.

## Histological analysis

Following overnight (O/N) fixation in 2% PFA at 4 °C, hearts were washed in PBS and dehydrated by passage through rising concentrations of ethanol. Samples were then washed in Butanol O/N, at RT, before being placed in molten 50:50 butanol: Histoplast paraffin wax (Fisher) at 56 °C. After 1 hr, the solution was replaced with 100% Histoplast paraffin wax. After several changes of 100% wax, hearts were oriented and embedded into a mould pre-loaded with paraffin. The wax was then rapidly cooled on a bed of ice and stored at 4 °C. A microtome was used to cut 10µm-thick sections through

the heart. Before staining, sections were first deparaffinised in Histoclear solution (Fisher), followed by rehydration through a decreasing concentration of ethanol. For Masson's Trichrome Staining, the Masson's Trichrome kit (Abcam) was used as per the manufacturer's instructions. Briefly, slides were immersed in Bouin's solution for 15 min at 56 °C. Sections were then stained by serial immersion in the following solutions: Weigert's Iron Haemotoxylin solution (5 min); Biebrich Scralet Acid Fuchsin (5 min); Aniline Blue (5 min); Phosphotungstic/Phosphomolybdic acid solution (5 min) and finally 1% Acetic acid solution (2 min). Samples were then rinsed in distilled water and dehydrated. Finally, sections were cleared in Xylene and mounted using DPX mounting media.

## Immunofluorescence staining

Embryos were harvested at the required embryonic stage, placed in ice-cold PBS (Sigma) and the heart micro-dissected. Similarly, hearts from neonates were removed and washed in ice-cold PBS. Both embryonic and neonatal hearts were fixed for 6 hr in 2% paraformaldehyde (PFA; Santa Cruz Biotechnology) at 4 °C and equilibrated in 30% sucrose overnight at 4 °C. Hearts were then placed in 50:50 30% sucrose/PBS: Tissue-Tek OCT (VWR) for 30 min at room temperature (RT) and embedded in OCT. 10-μm-thick cryosections through the heart were cut. Before use, slides were left to dry for 10 min at RT and then washed in PBS for 5 min to remove the OCT. Samples were permeabilised with 0.5% (sections) or 0.1% (explants) Triton X-100 in PBS (PBTr) for 10 min at RT and subsequently rinsed twice in PBS. Samples were blocked in 10% Serum, 1% bovine serum albumin (BSA, Merck), 0.1% PBTr for 1 hr at RT prior to incubation with the primary antibodies overnight at 4 °C. The following day, slides were washed three times for at least 5 min in 0.1% PBTr. Samples were incubated with Alexa Fluor-conjugated secondary antibodies (1:200 dilution; Invitrogen) and 4',6-Diamidino-2-Phenylindole, Dihydrochloride (DAPI; 0.1 μg/ml, Invitrogen), for 1 hr at RT, protected from light. After final washes, slides were mounted in 50% glycerol in PBS. For wholemount staining, samples were washed in 0.3% PBTr and blocked in 1% BSA (Merck), 0.3% PBTr for at least 2 hr. The samples were then incubated with primary antibodies in the blocking solution overnight at 4 °C. On the second day, the samples were washed at least five times in 0.3% PBTr and then incubated with secondary antibodies and DAPI (Invitrogen) diluted in PBS overnight at 4 °C. The samples were then washed with PBS at least five times the next day and mounted in 50% glycerol in PBS. The following primary antibodies were used: Dylight 549 Mab (Hypoxyprobe-1 Inc, 1:200) antibody to detect hypoxyprobe labelling, HIF-1α (1:100, Novus Biologicals), HIF-2α (1:100, R&D Systems), endomucin (1:50, Santa Cruz Biotech), podoplanin (1:200, Fitzgerald), WT1 (1:100, Abcam), Alexa Fluor 488 Phalloidin (1:250, ThermoFischer), actin alpha-smooth muscle Cy3 (1:100, Sigma), Anti-α-Actinin (Sarcomeric) (1:500, Sigma-Aldrich), CD31 (1:100, Abcam), smooth muscle Myosin heavy chain 11 (SM-MHC, 1:100, Abcam), SM22 alpha (1:100, Abcam), Wheat-germ agglutinin (WGA, 1:100, Thermofisher), prolyl hydroxylase domain-2 (PHD2, 1:100, Novus Biologicals). Endomucin staining of coronary vessels was quantified using AngioTool.

## RNA isolation and qRT-PCR analysis

Total RNA was isolated from frozen ventricles and epicardial explants with Trizol (Invitrogen) using a Teflon homogeniser followed by aspiration with a sterile 25-gauge needle and syringe (BD sciences). RNA was transcribed into cDNA utilising random primers (Promega) with Superscript Reverse Transcriptase III (Life Technologies). Real-time quantitative PCR was performed on a ViiA 7 Real-Time PCR System (Applied Biosystems), using SYBR Green mix (Invitrogen). Gene expression was evaluated as DeltaCt relative to control (*Atp5b*, *Sdha* and *B2m*).

## Heart dissociation and FACS sorting

Hearts from P1 and P7 mice were harvested and minced into a single cell suspension of FACS sorted live cells. Briefly, for each sample, 3 hearts collected from the same litter were pooled together and finely minced with a scalpel. Tissue was then digested by gentle agitation (180 rpm shaker) in Collagenase II (Gibco) using a solution of 500 units/ml in HBSS at 37 °C for 45 min (P7 hearts) or 20 min (P1 hearts). Cell solutions were then passed through a 70 μm filter, washed and incubated in Red Blood Cell lysis buffer (Cell Signaling Technology) for 10 min at RT to remove red blood cells. Finally, isolated single cardiac cells were centrifuged, resuspended in 2% FBS in PBS, passed through the filtering cap of the FACS tubes and incubated with 1% 7-AAD viability stain (Invitrogen) for 10 min. Approximately $1 \times 10^5$ live cells per sample were sorted using the BD FACSAria Fusion Sorter.

## Single cell RNA-sequencing

Hearts were isolated from intact P1 and P7 mice. FACS sorted cardiac cells' viability and concentration were assessed by using an automated cell counter and $1.5 \times 10^4$ cells per sample were loaded onto the 10 X Chromium system (10 X Genomics) to obtain a target cell recovery of ~6000 cells. Single cell RNA-seq libraries were generated using Single Cell 3 Prime Reagent Kits v1.3 (10x Genomics) according to the manufacturer's protocol. Sequencing was performed on an Illumina NovaSeq 6000 System operated by the Oxford Genomics Centre at the Wellcome Centre for Human Genetics. Raw sequence reads were aligned against the mouse mm10/GRCm38 reference transcriptome using the Cell Ranger 3.1.0 pipeline (10 x Genomics) and processed further using the scRNA-seq analysis R package Seurat (v.3.2.3). Initial filtering removed cells expressing less than 200 genes and genes that were expressed in less than 3 cells. To exclude low-quality cells and doublets, we filtered out cells with a very high mitochondrial gene content and cells that expressed more than 8000 genes. Based on these criteria, 19,211 genes across 5308 and 8734 cells for P1 and P7 samples respectively remained for downstream analysis. Data from the two samples were combined and scaled by regressing out the nUMIs, percentage of mitochondrial gene expression and cell cycle. To correct for batch effect, samples were integrated using the Harmony package (v.1.0). Uniform manifold approximation and projection (UMAP) was performed on the scRNAseq harmonised cell embeddings and unbiased clustering was obtained using the FindCluster function of the Seurat pipeline. Cluster cell types were annotated using a combination of differentially expressed markers, identified using the Seurat FindAll-Markers function and the expression of selected canonical markers for specific cell types. The epicardial cell cluster was then identified and subset into a new Seurat object with raw read counts. The standard Seurat pipeline described above was performed and the Model-based Analysis of Single-cell Transcriptomics test (MAST v.1.14.0) was used to analyse the differential gene expression between the P1 and P7 groups in the epicardial cells. Finally, differentially expressed genes were ranked based on both the fold change and p value (avg_logFC * -log10 of the p_val) and the ranked list was submitted to the Metascape platform (https://www.metascape.org) to identify enriched pathways in the P1 or P7 group, respectively.

## Cardiac cine-MRI

Cardiac cine-MRI was performed at 7T using a Varian DDR system. Briefly, mice were anaesthetised with 2% isoflurane in $O_2$ and positioned supine in a custom animal handling system with homeothermic control. Prospectively gated proton cardiac images were acquired with a partial Fourier accelerated spoiled gradient echo CINE sequence (TR 5.9ms, TE 2.2ms, 30 kHz bandwidth, 25° FA, approximately 20–30 frames gated to the R wave with a 4ms post-label delay; 20% partial acquisition; 4 averages) with a 72 mm volume transmit/4 channel surface receive coil (Rapid Biomedical GmbH) in order to acquire two and four chamber long-axis views and a short axis stack for functional quantification (128x128 matrix; 25.6 mm$^2$ FOV; 0.7 mm slice thickness, 0.2 mm resolution in-plane). Non-acquired partial Fourier data was reconstructed via the method of projection onto convex sets prior to a simple, cartesian, DFT. Blinded image analysis was performed in ImageJ as described previously. Mice with either excessive or minimal infarct sizes were excluded from further analyses by MRI scanning at day 1 post-MI.

## Wound healing/scratch assay

$2 \times 10^4$ MEC.1 cells were seeded on a two-well culture insert (Ibidi, Germany). The day after, the insert was removed and 50 µM of BAY 85–3934 (Molidustat, Selleck) or vehicle (DMSO) was added to the culture. Cells were imaged every 2 hr up to 6 hr. The cell-free space area was determined by using the ImageJ software (NIH, Rockville, USA).

## Image analysis

For quantification of immunofluorescence of cryosections, images were captured on an Olympus FluoView 3000 confocal microscope and analysed with Fiji (NIH) and AngioTool (*Zhou et al., 2011*). To quantify nuclear fluorescence, a macro was written to identify all DAPI-stained nuclei and to analyse nuclear fluorescence based on the DAPI mask. To localise WT1+ cells in the epicardium and myocardium, cells were manually counted. To quantify the cell size, three independent samples per

group with six different fields and positions, two from left and right ventricles, and septum were captured at ×40 magnification. ImageJ (National Institutes of Health) was used to quantify the size of each cell.

## Statistical analysis

All data are presented as mean ± SEM or as median, inter-quartile range (IQR) and upper and lower limits. Statistical analysis was performed on GraphPad Prism 8 software. The statistical significance between two groups was determined using an unpaired two-tailed Student's t-test, these included an F-test to confirm the two groups had equal variances. Among three or more groups, one-way ANOVA followed up by Dunnett's or Tukey's multiple comparison tests was used. A value of $p \leq 0.05$ was considered statistically significant.

## Acknowledgements

We are very grateful to Prof. Sir Peter J Ratcliffe for technical advice and highly informative scientific discussions throughout the project. We would like to thank Biomedical Services Unit for animal husbandry and Micron Oxford Advanced Bioimaging Unit (supported by Wellcome Strategic Awards 091911/B/10/Z and 107457/Z/15/Z) for access to and training in the use of confocal microscopy. This work was supported by the British Heart Foundation (BHF chair award to PRR: CH/11/1/28798; BHF programme grant to PRR: RG/18/33532; BHF Intermediate Basic Science Research Fellowship to JMV: FS/19/31/34158) and supported by the BHF Oxbridge Regenerative Medicine Centre (RM/17/2/33380).

## Additional information

### Funding

| Funder | Grant reference number | Author |
| --- | --- | --- |
| British Heart Foundation | RG/18/33532 | Paul R Riley<br>Elisabetta Gamen |
| British Heart Foundation | RM/17/2/33380 | Eleanor L Price |
| British Heart Foundation | FS/19/31/34158 | Joaquim Miguel Vieira |
| British Heart Foundation | CH/11/1/28798 | Paul R Riley<br>Carla De Villiers |

The funders had no role in study design, data collection and interpretation, or the decision to submit the work for publication.

### Author contributions

Elisabetta Gamen, Data curation, Formal analysis, Investigation, Methodology, Project administration, Writing – original draft; Eleanor L Price, Data curation, Formal analysis, Investigation, Methodology, Visualization; Daniela Pezzolla, Methodology, Data curation, Investigation, Project administration; Carla De Villiers, Mala Gunadasa-Rohling, Maria-Alexa Cosma, Judith Sayers, Carolina Roque Silva, Rafik Salama, David Robert Mole, Investigation; Adam B Lokman, Methodology, Data curation, Investigation; Tammie Bishop, Chris W Pugh, Robin P Choudhury, Resources; Carolyn A Carr, Methodology, Investigation; Joaquim Miguel Vieira, Methodology, Supervision, Project administration, Writing – review and editing; Paul R Riley, Conceptualization, Resources, Methodology, Supervision, Funding acquisition, Validation, Project administration, Writing – review and editing

### Author ORCIDs

Daniela Pezzolla ⓘ https://orcid.org/0000-0001-8060-3706
Adam B Lokman ⓘ https://orcid.org/0000-0002-8717-4550
Joaquim Miguel Vieira ⓘ https://orcid.org/0000-0003-2023-6304
Paul R Riley ⓘ https://orcid.org/0000-0002-9862-7332

### Ethics

All animal experiments were carried out according to UK Home Office project licences PPL PC013B246, PDDE89C84 and PP3194787 and were compliant with the UK Animals (Scientific Procedures) Act 1986. Mice were cared for and housed by Oxford University Biomedical Services. Mice were maintained in individually ventilated cages (IVCs) and ventilated racks at 22oC and 55% humidity. Neonatal mouse surgery was carried out under general anaesthesia, induced with 4 % isoflurane inhalation in oxygen and every effort was made to minimise suffering.

Reviewer #1 (Public review): https://doi.org/10.7554/eLife.107419.3.sa1
Reviewer #2 (Public review): https://doi.org/10.7554/eLife.107419.3.sa2
Reviewer #3 (Public review): https://doi.org/10.7554/eLife.107419.3.sa3
Author response https://doi.org/10.7554/eLife.107419.3.sa4

## Additional files

### Supplementary files

MDAR checklist

### Data availability

The scRNA-Seq dataset analysed herein has been uploaded into the Gene Expression Omnibus (GEO) repository, accession number GSE305621.

The following dataset was generated:

| Author(s) | Year | Dataset title | Dataset URL | Database and Identifier |
|---|---|---|---|---|
| Choudhury RP, Pezzolla D, Mohan K, Riley PR | 2025 | Single-cell RNA-seq dataset of P1 and P7 mice hearts | https://www.ncbi.nlm.nih.gov/geo/query/acc.cgi?acc=GSE305621 | NCBI Gene Expression Omnibus, GSE305621 |

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
