## [Editor Report · eLife Assessment]

This **valuable** study investigates the role of HIF1a signaling in epicardial activation and neonatal heart regeneration in mice. Using a combination of genetic and pharmacological approaches, the authors demonstrate that stabilization of HIF1a enhances epicardial activation and extends the regenerative capacity of the heart beyond the typical neonatal window following myocardial infarction. The main conclusion is well supported by **solid** data, although some minor concerns regarding experimental interpretation require further clarification to ensure accuracy.

---

## [Referee Report · Reviewer #1 (Public review)]

Summary:

The manuscript by Gamen et al. analyzed the functional role of HIF signaling in the epicardium providing evidence that stabilization of the hypoxia signaling pathway might contribute to neonatal heart regeneration. By generating different conditionally mouse mutants and performing pharmacological interventions, the authors demonstrate that stabilizing HIF signaling enhances cardiac regeneration after MI in P7 neonatal hearts.

Strengths:

The study presents convincing genetic and pharmacological approaches on the role of hypoxia signaling enhance the regenerative potential of the epicardium

Weaknesses:

The major weakness remains the lack of convincing evidence demonstrating the role of hypoxia signaling in EMT modulation in the epicardial cells. The authors claimed that EMT assays adopted in this study are based on similar previous studies. Surprisingly, two of the references provided correspond to their own research group (PMID: 17108969, PMID: 19235142), limiting the credit for such claims, and the other two (PMID: 27023710, PMID: 12297106) assessment of cell migration but not EMT is reported. Thus, EMT remains to be convincingly demonstrated.

---

## [Referee Report · Reviewer #2 (Public review)]

Summary:

In this study, Gamen et al. investigated the roles of hypoxia and HIF1a signaling in regulating epicardial function during cardiac development and neonatal heart regeneration. The authors identified hypoxic regions in the epicardium during development and demonstrated that genetic and pharmacological stabilization of HIF1a during neonatal heart injury prolonged epicardial activation, preserved myocardium, enhanced infarct resolution, and maintained cardiac function beyond the normal postnatal regenerative window.

Strengths:

HIF1a signaling was manipulated in an epicardium-specific manner using appropriate genetic tools.

Weaknesses:

Some conclusions still need clarification.

Comments on revisions:

(1) The authors' comment on the partial overlap of HP1 and HIF1a IF signals (HIF1a is highly unstable ... broader regions of hypoxia) is reasonable and would help readers interpret the data if included in the text describing Fig. 1.

(2) The conclusion regarding WT1+ cells in Fig. 2a and b remains unclear. Both panels display larger and smaller magenta cells, and when all are taken into account, the overall number does not appear substantially different. Additional clarification is needed on how the quantification was performed.

(3) Regarding Figure 6-figure supplement 1c, it seems difficult to conclude the endothelial identity of WT1+ cells based on EMCN staining, as the markers do not overlap. The authors note that WT1 is upregulated in endothelial cells, but this has been reported in the context of injury, which differs from the context of the present study involving Molidustat.

---

## [Referee Report · Reviewer #3 (Public review)]

Summary:

The author's research here was to understand the role of hypoxia and hypoxia-induced transcription factors Hif-1a in the epicardium. The authors noted that hypoxia was prevalent in the embryonic heart and this persisted into neonatal stages until post natal day 7 (P7). Hypoxic regions in the heart were noted in the outer layer of the heart and expression of Hif-1a coincided with the epicardial gene WT1. It has been documented that at P7, the mouse heart cannot regenerate after myocardial infarction and the authors speculated that the change in epicardial hypoxic conditions could play a role in regeneration. The authors then used genetic and pharmacological tools to increase the activity of Hif genes in the heart and noted that there was a significant improvement in cardiac function when Hif-1a was active in the epicardium. The authors speculated that the presence of Hif-1a improved cell survival.

Strengths:

A focus on hypoxia and its effects on the epicardium in development and after myocardial infraction. This study outlines a potential to extend the regenerative time window in neonatal mammalian hearts.

Weaknesses:

While the observations of improved cardiac function is clear, the exact mechanism of how increased Hif-1a activity causes these effects is not completely revealed. The authors mention improved myocardium survival, but do not include studies to demonstrate this.

There is an indication that fibrosis is decreased in hearts where Hif activity is prolonged, but there are no studies to link hypoxia and fibrosis.

Comments on revisions:

In the manuscript revision, the authors address my comments. They outline differences between genetic disruption of Phd2 and chemical inactivation could be due to dosing and drug half-life of Molidustat. The other comments are addressed by explaining that they have analyzed enough heart sections and hearts to come to their conclusions. The authors also state they cannot generate more numbers for this study, therefore I accept their conclusions as stated.

---

## [Author Response]

The following is the authors’ response to the original reviews

**eLife Assessment**
This valuable study investigates the role of HIF1a signalling in epicardial activation and neonatal heart regeneration in mice. Through a combination of genetic and pharmacological approaches, the authors show that stabilization of HIF1a enhances epicardial activation and extends the regenerative capacity of the heart beyond the typical neonatal window following myocardial infarction (MI). However, several aspects of the study remain incomplete and would benefit from further clarification and additional experimental support to solidify the conclusions.

We reveal herein prolonged epicardial activation following myocardial infarction (MI) beyond post-natal days 1-7 (P1-P7) by genetic or pharmacological stabilisation of HIF-signalling. This extends the so-called “regenerative window” during an adult-like response to injury, leading to enhanced survived myocardium and functional improvement of the heart, even against a backdrop of persistent, albeit reduced, fibrosis. The epicardium is known to enhance cardiomyocyte proliferation and myocardial growth during heart development via trophic growth factor (for example, IGF-1, FGF, VEGF, TGFβ and BMP) signalling (reviewed in PMID:29592950) and epicardium-derived cell-conditioned medium reduces infarct size and improves heart function (PMID: 21505261). Further experiments, outside of the scope of the current study, are required to determine whether activated neonatal epicardium elicits similar paracrine support to sustain the myocardium and heart function after injury beyond P7 into adulthood.

**Public Reviews:**

**Reviewer #1 (Public review):**
Summary:The manuscript by Gamen et al. analyzed the functional role of HIF signaling in the epicardium, providing evidence that stabilization of the hypoxia signaling pathway might contribute to neonatal heart regeneration. By generating different conditionally mouse mutants and performing pharmacological interventions, the authors demonstrate that stabilizing HIF signaling enhances cardiac regeneration after MI in P7 neonatal hearts.Strengths:The study presents convincing genetic and pharmacological approaches to the role of hypoxia signaling in enhancing the regenerative potential of the epicardium.Weaknesses:The major weakness is the lack of convincing evidence demonstrating the role of hypoxia signaling in EMT modulation in epicardial cells. Additionally, novel experimental approaches should be performed to allow for the translation of these findings to the clinical arena.

We respectfully disagree that we have not convincingly demonstrated a role for HIF-signalling in promoting epicardial EMT. We adopt epicardial explant assays utilising a well characterised ex vivo protocol previously described for studying EMT in embryonic, neonatal and adult epicardium (PMID: 27023710, PMID: 12297106; PMID: 17108969, PMID: 19235142). These assays demonstrate in WT1^CreERT2^;Phd2^fl/fl^ explants enhanced cobblestone to spindle-like change in cell morphology, increased cell migration, appearance of stress fibres and an up-regulation of the mesenchymal marker alpha-smooth muscle actin (αSMA); all parameters associated with EMT. In addition, our in vivo analyses of Wt1^CreERT2^;Phd2^fl/fl^ hearts, in response to neonatal injury, reveal elevated numbers of WT1+ epicardial cells within the sub-epicardial region and underlying myocardium as is associated with active EMT and subsequent migration from the epicardium.

**Reviewer #2 (Public review):**
Summary:In this study, Gamen et al. investigated the roles of hypoxia and HIF1a signaling in regulating epicardial function during cardiac development and neonatal heart regeneration. They found that WT1^+^ epicardial cells become hypoxic and begin expressing HIF1a from mid-gestation onward. During development, epicardial HIF1a signaling regulates WT1 expression and promotes coronary vasculature formation. In the postnatal heart, genetic and pharmacological upregulation of HIF1a sustained epicardial activation and improved regenerative outcomes.Strengths:HIF1a signaling was manipulated in an epicardium-specific manner using appropriate genetic tools.Weaknesses:There appears to be a discrepancy between some of the conclusions and the provided histological data. Additionally, the study does not offer mechanistic insight into the functional recovery observed.

We respectfully disagree with the comment that our histological data does not support our conclusions and expand on this in the response to specific reviewer comments. We agree that further mechanistic experiments outside of the scope of the current study are required to identify precisely how activated neonatal epicardium results in increased healthy myocardium after injury beyond post-natal day 7 (P7).

**Reviewer #3 (Public review):**
Summary:The authors' research here was to understand the role of hypoxia and hypoxia-induced transcription factor Hif-1a in the epicardium. The authors noted that hypoxia was prevalent in the embryonic heart, and this persisted into neonatal stages until postnatal day 7 (P7). Hypoxic regions in the heart were noted in the outer layer of the heart, and expression of Hif-1a coincided with the epicardial gene WT1. It has been documented that at P7, the mouse heart cannot regenerate after myocardial infarction, and the authors speculated that the change in epicardial hypoxic conditions could play a role in regeneration. The authors then used genetic and pharmacological tools to increase the activity of Hif genes in the heart and noted that there was a significant improvement in cardiac function when Hif-1a was active in the epicardium. The authors speculated that the presence of Hif-1a improved cell survival.Strengths:A focus on hypoxia and its effects on the epicardium in development and after myocardial infarction. This study outlines the potential to extend the regenerative time window in neonatal mammalian hearts.

We thank the reviewer for this positive endorsement and recognition of the importance of mechanistic insight into how to extend the window of neonatal heart regeneration.

Weaknesses:While the observations of improved cardiac function are clear, the exact mechanism of how increased Hif-1a activity causes these effects is not completely revealed. The authors mention improved myocardium survival, but do not include studies to demonstrate this.

We report an increase in healthy myocardium arising from prolonged activation of the epicardium during the neonatal window and following injury at post-natal day 7 (P7). We speculate this recapitulates the role of the epicardium during heart development which is known to be a source of trophic growth factors that can enhance myocardial growth. Further experiments are required, out-of-scope of this study, to define a mechanistic link between HIF-signalling, epicardial activation and myocardial survival in the setting of prolonged neonatal heart regeneration.

There is an indication that fibrosis is decreased in hearts where Hif activity is prolonged, but there are no studies to link hypoxia and fibrosis.

We believe the decreased fibrosis is a natural consequence of the increase in survived myocardium arising from the activated epicardium. There is strong precedent here following injury at post-natal day 1 (P1) in which fibrosis is evident early-on but is resolved over time with growth of the myocardium in the regenerating heart (PMID: 23248315).

**Recommendations for the authors:**

**Reviewing Editor Comments:**
(1) Address issues related to image quality, colocalization, sample labeling, appropriate controls, and quantification - particularly in Figures 1, 2, 6, and Supplementary Figure 9. Increase sample size as noted by reviewers.

The issues of co-localisation and sample labelling have been addressed under response to reviewers. We are unable to increase sample numbers but have clarified the number of regions per section and numbers of sections per heart analysed where appropriate.

(2) Clarify the effects of epicardial HIF1a activation on neovascularization.

We have removed reference in the abstract to an effect on neovascularisation.

(3) Extend assessments of epicardial hypoxia and HIF1a expression to earlier embryonic stages, when epicardial EMT is more active.

Our earliest timepoint of E12.5 marks the onset of epicardial EMT and E13.5 is the stage with the most significant mobilisation of epicardium-derived cells (EPDCs) into the sub-epicardial region and underlying myocardium (PMID: 32359445). In the same study, E11.5 lineage tracing of epicardial cells is restricted to outer layer of the heart; thus, our timepoints are representative in capturing both the onset and progression of in vivo EMT.

(4) Strengthen EMT assays and mechanistic modeling. Provide evidence from physiologically relevant models, as current 2D culture assays do not adequately support conclusions about EMT. Include additional EMT markers and quantification where appropriate.

We respectfully disagree that epicardial explants are not a valid assay for assessing EMT. As noted under responses to reviewers, such primary explants have been widely described elsewhere (PMID: 27023710, PMID: 12297106; PMID: 17108969, PMID: 19235142) and enable documentation of multiple parameters that are associated with active EMT, including an assessment of the extent of cell migration, cobblestone (epithelial) to spindle-like (mesenchymal) cell morphologies, stress fibre formation and expression of alpha-smooth muscle actin as a mesenchymal marker. We support our findings in explants by revealing reduced WT1+ epicardium-derived cells (EPDCs) in the sub-epicardial region and underlying myocardium of WT1^CreERT2/+^;Hif1a^fl/fl^ embryonic hearts (data in Figure 2) indicative of impaired epicardial EMT and migration of EPDCs and in vivo following neonatal MI with pharmacological inhibition of PHD2, where we observe the reciprocal phenotype of increased numbers of epicardium-derived cells emerging from the outer epicardial layer (data in Figure 6).

(5) Strengthen mechanistic insights into the role of epicardial cells in the functional recovery observed in MI hearts.

We agree that further experiments are required, out-of-scope of this study, to define a mechanistic link between HIF-signalling, epicardial activation and myocardial survival in the setting of prolonged neonatal heart regeneration.

**Reviewer #1 (Recommendations for the authors):**
The manuscript by Gamen et al. analyzed the functional role of HIF signaling in the epicardium, providing evidence that stabilization of the hypoxia signaling pathway might contribute to neonatal heart regeneration. By generating different conditionally mouse mutants and performing pharmacological interventions, the authors demonstrate that stabilizing HIF signaling enhances cardiac regeneration after MI in P7 neonatal hearts. The study is potentially interesting, but it presents several major caveats.(1) One of the critical points reported in the early stages of this study is the early co-localization of Wt1, the hypoxic report (HP1), and HIF signaling pathways master regulators (i.e., HIF1a and HIF1b) during embryonic development. Figure 1 is meant to report such findings. However, unfortunately, I hardly see any co-localization at all in the Wt1+ epicardial cells for HP1, with some colocalization is seen for HIF1 and 2 alpha, although none of these data are quantified. Thus, it is hard to believe such co-localization.

We respectfully disagree with this comment. We highlight cells in Figure 1 that are co-stained for WT1+ and HP1. In addition, we identify HIF1-α and HIF2- α positive cells which either reside within the epicardium, as the outer cell layer, or within the underlying sub-epicardial region, respectfully.

(2) The authors claimed that they have analyzed the expression of the hypoxic report, as well as Wt1 and the HIF signaling pathways master regulators (i.e., HIF1a and HIF1b) in the AV groove, as compared to the apex, in embryonic heart ranging from E12.5 to E18.5 (Figure 1). Unfortunately, all images provided that are tagged as AV groove are rather misleading. They do not represent the AV groove but part of the right ventricular free wall. If the authors want to refer to the AV groove, AV cushions should be visible underneath.

We have removed specific reference to the AV groove and refer to the highlighted regions as the “Base” of the heart.

(3) The authors analyzed the hypoxic condition of the developing heart from E12.5 to E18.5. However, it remains unclear why the authors only explored the hypoxic conditions from E12.5 onwards, since epicardial EMT mainly occurs earlier than this time point, i.e., E10.5 onwards. Therefore, it would be needed to explore it already at this earlier time point.

We respectfully disagree with the reviewer and refer to the comment above regarding the fact that E12.5 marks the onset of epicardial EMT and E13.5 is the stage with the most significant mobilisation of epicardium-derived cells (EPDCs) into the sub-epicardial region and underlying myocardium (PMID: 32359445).

(4) The authors reported a conditional mouse model of HIF1alpha deletion by using the Wt1CreERT2 driver. Curiously, Wt1 is dependent on hypoxia signaling (i.e., HIF1a). Therefore, it is unclear whether there is a negative feedback loop between the deletion of Hif1alpha and the activation of the Cre driver might have functional consequences. Convincing evidence should be provided that such crosstalk does not interfere with Hif1alpha inactivation, and therefore, appropriate controls should be run in parallel.

We discount a negative feedback loop in this instance based on the fact we have utilised heterozygous mice for the WT1^CreERT2/+^ line and observe a consistent and reproducible phenotype for the developing hearts on a Wt1^CreERT2/+^;Hif1a^fl/fl^ background and following injury in Wt1^CreERT2/+^;Phd2^fl/fl^ mice. Collectively this indicates that the WT1-CreERT2 driver is active in the context of diminishing HIF-1α and Phd2, respectively. In addition, have carried out parallel experiments using epicardial explants derived from R26R-CreERT2;Phd2^fl/fl^ (Figure 3) to circumvent any potential confounding issues; the results of which are consistent with increased epicardial EMT in support of our overall hypothesis.

(5) On Figure 2a-f the authors reported that epicardial cells are diminished in Wt1CreERT2Hif1alpha mice as compared to controls. I am very sorry, but I do not see any difference. Furthermore, it is unclear to me how the authors quantified such differences, i.e., what marker signal did they use and how it was performed (Figure 2c and d)?

We respectfully disagree with the reviewer and draw attention to the single channel panels of WT1+ staining in Figure 2, which show clear differences between numbers of epicardial cells in the mutant mice compared to controls (comparing magenta cells in panels a) versus (b). Quantification was carried out for numbers of WT1+ cells residing within the PDPN-positive epicardium (and underlying PDPN-negative myocardium) across multiple images from multiple sections and multiple hearts.

(6) On Figure 2g, the authors reported differences in total vessel length. Are they referring to impaired microvasculature development? Or is this analysis also including major coronary vessels? What about the major coronary vessels and trees, is there any affection?

This analysis refers to the microvasculature and not the major coronary arteries or coronary trees.

(7) The authors reported that there might be some differences in EMT markers, but unfortunately, all of them are analyzed on 2D cultures, where no substrate for EMT is present, i.e., an underlying ECM bed. Thus, the authors cannot claim that EMT is altered. Additional experiments using either collagen substrate and/or Matrigel are required to fully demonstrate that EMT is impaired. Furthermore, quantitative analyses of such differences should be provided.

The 2D cultures are epicardial explants from mutant versus wild type hearts and represent a widely adopted previously published ex-vivo assay for investigating epicardial EMT across embryonic to adult stages (PMID: 27023710, PMID: 12297106; PMID: 17108969, PMID: 19235142); including an assessment of the extent of migration and cobblestone (epithelial) to spindle-like (mesenchymal) cell morphologies, stress fibre formation and expression of alpha-smooth muscle actin as a mesenchymal marker. We do not understand the comment regarding an “underlying ECM bed” as the cells exhibit EMT routinely on tissue culture plastic and will deposit their own ECM during the culture time course and in response to EMT/cell migration. In terms of quantification this was carried out for scratch assay experiments, as a proxy for EMT and emergent mesenchymal cell migration, as presented in Figure 3i, j with significant enhanced scratch closure and cell migration following Molidustat treatment.

(8) The description of data provided on Supplementary Figure 5 is spurious and should be removed. A note in the discussion might be sufficient.

We respectfully disagree. The ChIP-seq data, in what is now Figure 2- figure supplement 3, highlights a HIF-1 α binding site within the *Wt1* locus suggesting putative upstream regulation of WT1 by HIF-1α. Thus this provides a potential explanation as to how HIF-1α may activate the epicardium through up-regulation of *Wt1*/WT1.

(9) On Figure 3, the authors further illustrate the change of EMT markers using ex vivo cardiac explants. They reported increased expression of Snai2 that, although statistically significant, is most likely of no biological relevance (increase of only 20% at transcript level). What about Snai1, Prrx1, and other EMT promoters? Are they also induced? As previously stated, these 2D cultures do not provide supporting evidence that EMT is occurring, thus 3D gel assays should be performed in which Z-axis analyses will provide evidence on the different migratory behaviour of those cells.

We respectfully suggest that a 20% change in snai2 expression is biologically meaningful with respect to EMT. This in-turn is supported by associated cell migration, reduced ZO-1 expression, increased stress fibres and increased alpha-SMA as a mesenchymal marker; all properties associated with active EMT. Other suggested markers have not been validated as formally required for EMT, for example Snai1 (PMID: 23097346). The migratory capacity of targeted versus epicardial cells was assessed by combined explant and scratch assay experiments.

(10) The description of single-cell analyses is very incomplete. Which mice were used for these analyses, wildtype control, or hypoxic mice? Please provide a clearer description of the samples used. Additionally, the entire rationale of these analyses is dubious. Doing single-cell analyses to analyze a couple or three markers in a very small cell population is rather ridiculous. qPCR might be far more appropriate and convincing, or a bulk RNAseq analysis of isolated epicardial cells.

The single-cell analyses represent an unbiased assessment of different pathways in epicardial cells (identified bioinformatically) between intact P1 and P7 stages in wild type (control) hearts, with a focus on hypoxia-related gene expression and HIF-dependent pathways. It was not designed to analyse a small number of genes, rather global differences in the hypoxic states between P1 and P7 hearts. Selected genes (Vegfa, Pdk3, Egln 1 (Phd2)) were analysed to highlight the key differences in hypoxic signalling across the regenerative window. The fact the hearts were uninjured/intact is clarified in the text and legends for Figure 4 and now Figure 4-figure supplement 1.

(11) The analyses provided in Figure 5 are very interesting and their findings are very relevant. However, I would think that the complementary experimental approach should also be done, i.e, MI followed by activation with tamoxifen, since that situation would be more realistic in the clinical setting.

Tamoxifen causes respiratory failure in neonates with MI, so the two cannot be combined at the same time or soon after surgery. Moreover, tamoxifen takes significant time to take effect on targeted gene down-regulation which may negate sufficient activation of the epicardium following injury.

The experiments in Figure 5 were designed to demonstrate that prolonged heart regeneration could be elicited in a cell-specific (epicardial-specific) manner via a genetic approach. The pharmacological experiments in Figure 6 are complementary in this regard by demonstrating equivalent effects with drug (Molidustat) delivery to reduce PHD2 and stabilise HIF post-MI.

(12) In Figure 6, expression of Wt1 is highly prominent in P7 controls, mainly restricted to the epicardial lining while in the experimental setting, such Wt1 expression is broadly distributed on the subepicardial space, nicely demonstrating epicardial activation. However, it is very surprising to see such Wt1 expression in controls, something that is not expected, as compared to the data reported in Figure 4g. Could the authors please reconcile these findings?

Figure 6 represents the injury setting and Figure 4g the intact setting (as clarified above, in the text and revised figure legends). Hence in the latter WT1 expression is significantly reduced in the P7 heart, as anticipated. With injury at P7 we anticipate activation of WT1 in control hearts, albeit restricted to the epicardial layer (as occurs in adult hearts, PMID: 21505261). In contrast, following Molidustat-treatment of P7 hearts post-MI we observe extensive epicardial expansion into the sub-epicardial region and EPDC migration into the underlying myocardium (Figure 6b).

**Reviewer #2 (Recommendations for the authors):**
The role of hypoxia and HIF1a signaling in epicardial activation is an important topic, and the genetic approaches employed in this study are appropriate. However, several aspects of the study remain unclear and would benefit from further clarification or explanation by the authors:(1) The authors detected hypoxic regions using an anti-pimonidazole fluorescence-conjugated monoclonal antibody (HP1). The data would become more compelling if negative and positive controls were provided.

We believe the HP1 staining is compelling in the images shown and is consistent with hypoxic regions of the developing heart. We reveal HP1 staining at cellular resolution with neighbouring cells positive and negative for the HP1 signal in the apex of the heart and within the epicardium and sub-epicardial regions at E12.5 (Figure 1a) and diminished/altered hypoxic/HP1 regional signal through subsequent developmental stages at E14.5-18.5 (Figure 1a-d).

(2) Many HIF1a-positive cells in the AV groove region do not appear to overlap with HP1 staining (Figure 1a). Providing a low-magnification image of HIF1α expression would be helpful to better assess the extent of overlap with HP1 staining

HIF-1α is highly unstable and hence detection of HIF-1α+ cells will likely only sample of cells compared to HP1 which is a surrogate for broader regions of hypoxia.

(3) Although the authors conclude that epicardial HIF1a deletion results in a significant reduction of WT1⁺ cells in both the epicardium and myocardium (Figure 2a-d), the provided images are not sufficiently clear to fully support this interpretation. Providing additional evidence to support this conclusion would be helpful.

We respectfully disagree with the reviewer and draw attention to the single channel panels of WT1+ staining which show clear differences between numbers of epicardial cells in the mutant mice compared to controls (Figure 2a versus 2b; magenta WT1+ staining).

(4) Similar to the point raised above, the authors' conclusion regarding the increased expression of WT1 following Molidustat treatment does not appear to be fully supported by the provided images (Figure 6b-f). Immunofluorescence staining for WT1 does not clearly demonstrate epicardial expression in the remote zone of either the control or Molidustat-treated hearts. In addition, while an increase of WT1^+^ cells is observed in the infarct zone of the Molidustat-treated heart, it is somewhat unexpected that such expansion is not evident in the corresponding region of the control heart, given that epicardial cells typically expand near the infarct area. Clarification on these points would be helpful.

Figure 6b reveals WT1 expression in controls (upper panel set) that is reactivated proximal to the infarct region, given WT1 is not expressed in adult epicardium but restricted to the epicardial layer (as occurs in injured adult mouse hearts PMID: 21505261). This contrasts with what is observed in the Molidustat-treated P7 hearts post-MI, where we observe epicardial expansion and migration of WT1+ cells into the underlying myocardium (Figure 6b, lower panel set, infarct zone).

(5) The authors conclude that WT1^+^ cells in the myocardial tissue exhibit endothelial identity based on the colocalization of WT1 and EMCN signals (Supplementary Figure 9c). However, this interpretation is difficult to assess, as WT1 is a nuclear marker and EMCN is a membrane protein, which makes precise colocalization challenging to confirm with confidence. Additional supporting evidence may be necessary to substantiate this conclusion.

WT1 is known to be up regulated in endothelial cells in response to injury as shown previously in several studies (for example, PMID: 25681586). Here we show clear co-localisation of nuclear WT1 and cytoplasmic Endomucin (EMCN) in what is now Figure 6- figure supplement 1c and would encourage the reviewer and readers to magnify the image by zooming-in on the relevant co-stained panel.

(6) The authors conclude that activation of epicardial HIF1a signaling has no effect on neovascularization in postnatal MI hearts (Figure 5c). However, the abstract states: "Finally, a combination of genetic and pharmacological stabilisation of HIF ... increased vascularisation, augmented infarct resolution and preserved function beyond the 7-day regenerative window" (Lines 38-41). Clarification regarding this apparent discrepancy would be appreciated.

The abstract has been altered to remove the statement of increased vascularisation.

(7) The study appears somewhat incomplete, as it lacks mechanistic insight into the functional recovery observed following epicardial Phd2 deletion and Molidustat treatment in postnatal MI hearts. Although the authors suggest a potential paracrine role of the epicardium in protecting cardiomyocytes from apoptosis, this hypothesis has not been experimentally addressed. Incorporating such analysis would help to reinforce the study's conclusions.

Further experiments are required, which are out-of-scope of this study, to define a mechanistic link between the genetic or pharmacological stabilisation of HIF-signalling, epicardial activation and myocardial survival in the setting of prolonged neonatal heart regeneration.

Other points:(1) Providing single-channel images for Figures 1a-d and 6g would be helpful for clarity and interpretation.

We believe the combined channel views of co-staining for two markers on a background of DAPI staining to pin-point cell nuclei, are informative and support our conclusions.

(2) Have the authors considered using AngioTool to quantify the number of vessels in Figure 5b-c?

AngioToolTM was used to quantify the vessels, as we have used previously (PMID: 33462113) and this is now added to the methods and legend of Figure 2.

**Reviewer #3 (Recommendations for the authors):**
There are several areas where the manuscript can be improved, such that its conclusions can be solidified.(1) The authors highlight a point where blocking Phd2 can enhance survival of cardiac tissue, but did not report on survival markers. They surmised that apoptosis could be decreased in Phd2 mutant or Molidustat treatment but did not show this. The authors should determine if apoptosis is decreased in the myocardium and epicardium.

We show evidence of increased levels of healthy myocardium in the genetic and pharmacological models of stabilised HIF-signalling. We exclude increased cardiac hypertrophy or increased cardiomyocyte proliferation as causative, so suggest as a reasonable alternative enhanced survival, albeit this need not necessarily be via an apoptotic pathway given the incidence of necrotic cell death during MI. We are unable to generate new surgeries and mutant/treated heart samples to analyse for apoptotic markers at this stage.

(2) There appears to be no difference in cardiomyocyte proliferation in Molidustat-treated animals, but the experiment was only performed on 2 to 3 animals. This is too small a sample size to conclude from these results. The authors should increase the sample size to make this assertion.

We respectfully disagree that we are unable to conclude no effect on cardiomyocyte proliferation. We analysed multiple heart regions per section, for EdU+/cTnT+ colocalised signals across several sections per heart, set against a consistency of effect on other parameters in hearts treated with Molidustat. We are unable to generate more P7 heart surgeries +/- Molidustat and +/- EdU at this stage.

(3) It is curious as to how, after myocardial infarction, the fibrotic scar tissue is decreased in the Phd2 deletion but not as profound in Molidustat-treated mice at d21. Can the authors speculate why the difference exists and how this decrease arises? For example, are there decreased pro-inflammatory signals in Phd2 deleted mice? Is there decreased collagen deposition and ECM gene expression? Do macrophage recruitment into the infarct zone differ between mutant/treated vs WT?

The representative images in Figure 6k reveal a trend towards reduced fibrosis with Molidistat treatment (Figure 6l), but across all hearts analysed this was not as significant as observed in the epicardial-specific deletion injured hearts (Figure 5g, h). This may be due to the relatively short half-life of Molidustat (approximately 4-10 hours, PMID: 32248614), the dosing regimen for the drug and/or the fact that it was not specifically delivered/targeted to the epicardium.

(4) The magnified images in Figure 1 do not match the boxes in the whole heart images. It is unclear what the white boxes signify.

The white boxes have been removed from Figure 1. The magnified image panels are from serial heart sections and this is now clarified in the Figure 1 legend.